# YAP1/TAZ-TEAD transcriptional networks maintain skin homeostasis by regulating cell proliferation and limiting KLF4 activity

Yao Yuan[1,2], Jeannie Park [1], Amber Feng[1], Parirokh Awasthi[3], Zhiyong Wang[2,4], Qianming Chen[2] & Ramiro Iglesias-Bartolome [1✉]

The Hippo TEAD-transcriptional regulators YAP1 and TAZ are central for cell renewal and cancer growth; however, the specific downstream gene networks involved in their activity are not completely understood. Here we introduce TEADi, a genetically encoded inhibitor of the interaction of YAP1 and TAZ with TEAD, as a tool to characterize the transcriptional networks and biological effects regulated by TEAD transcription factors. Blockage of TEAD activity by TEADi in human keratinocytes and mouse skin leads to reduced proliferation and rapid activation of differentiation programs. Analysis of gene networks affected by TEADi and YAP1/TAZ knockdown identifies KLF4 as a central transcriptional node regulated by YAP1/TAZ-TEAD in keratinocyte differentiation. Moreover, we show that TEAD and KLF4 can regulate the activity of each other, indicating that these factors are part of a transcriptional regulatory loop. Our study establishes TEADi as a resource for studying YAP1/TAZ-TEAD dependent effects.

[1] Laboratory of Cellular and Molecular Biology, Center for Cancer Research, National Cancer Institute, National Institutes of Health, Bethesda, MD, USA. [2] State Key Laboratory of Oral Diseases, National Clinical Research Center for Oral Diseases, West China Hospital of Stomatology, Sichuan University, Chengdu, China. [3] Laboratory of Animal Sciences Program, Leidos Biomedical Research Inc., Frederick National Laboratory for Cancer Research, National Institutes of Health, Frederick, MD, USA. [4] Moores Cancer Center, University of California, San Diego, La Jolla, CA, USA. ✉email: ramiro.iglesias-bartolome@nih.gov

The barrier function of the skin is heavily dependent on the balance between self-renewal and differentiation of its basal stem/progenitor cell population. Transitions between stemness and differentiation states are determined by transcriptional and epigenetic changes that shape the genomic landscape of epithelial cells, affecting the differential binding of transcriptional activators and repressors to genes that participate in stem cell-renewal and the initiation of terminal differentiation[1–3]. YAP1 and its paralog TAZ (WWTR1) are co-transcriptional regulators downstream of the Hippo pathway[4,5] that are essential for skin homeostasis and epithelial stem cell maintenance[6–9]. YAP1 and TAZ are also implicated in skin basal (BCC) and squamous (SCC) cell carcinoma formation[7,10,11]. Indeed, this axis has been recently listed as one of the top 10 signaling pathways altered in human cancer[12].

The main stem cell regulatory and oncogenic functions of YAP1 and TAZ have been attributed to their interaction with TEAD transcription factors[9,13–15]. TEAD transcriptional-networks are a main component of the initiation of the epithelial state and are downregulated during keratinocyte differentiation[2], highlighting the relevance of TEADs in epithelial homeostasis. However, studies of the role of YAP1/TAZ in skin homeostasis and cancer involve the knockout of these co-transcriptional activators, impinging not only on TEAD-dependent events, but also potentially affecting a myriad of other transcriptional and signaling components that interact with YAP1 and TAZ[16,17]. For example, YAP1 and TAZ have been proposed to regulate Wnt signaling by interacting with the destruction complex[18] and by sequestering disheveled in the cytoplasm[19]. The disruption of these numerous cytoplasmic and nuclear pathways makes it difficult to pinpoint exactly the TEAD-specific effects mediated by YAP1 and TAZ, particularly in vivo.

To characterize the precise transcriptional events regulated downstream of YAP1/TAZ-TEAD, we introduce TEADi (TEAD-inhibitor), a genetically encoded fluorescently-tagged inhibitor of the interaction of YAP1 and TAZ with TEAD transcription factors. Here we use TEADi to identify the transcriptional networks regulated by TEAD using as a model human keratinocytes and mouse skin.

## Results

**Development of TEADi.** To target TEAD transcriptional activity we designed a genetically encoded dominant-negative protein (TEADi) containing TEAD-interacting domains that bind to TEAD and prevent its interaction with co-transcriptional activators. Specific TEAD-interacting domains are present in the Hippo-family members YAP1 and TAZ, and in vestigial-like (VGLL) proteins[20,21], and it has been shown that peptides containing these domains are effective in antagonizing YAP1 activity by blocking YAP1 binding to TEAD[21,22].

TEADi was constructed using as a starting point the TEAD-binding-domain (TBD) of the VGLL4 protein (TBD VGLL4) and the TEAD binding domain of YAP1 (TBD YAP1), previously shown to cooperatively block the interaction between YAP1 and TEADs as part of the Super-TDU peptide[21]. This sequence was modified by including a flexible linker sequence and by introducing a Pro98Glu substitution in TBD YAP1 (TBD YAP1*) that improves binding affinity to TEAD[20]. Genetically-encoded inhibitors give more flexibility to increases in protein size compared with soluble peptide inhibitors, allowing us to include the TEAD interaction domain of TAZ (TBD TAZ)[20] to increase inhibition towards TAZ, a bipartite nuclear localization signal (BPNLS)[23] to target the construct, and a green fluorescent protein (GFP) to easily track expression and localization of TEADi. The final construct consists of the following domains

separated by linker sequences: GFP-TBD VGLL4-TBD YAP1*-TBD TAZ-BPNLS (Fig. 1a, see full sequence details in the Methods section). TEADi has a predicted molecular weight of 39 kDa, its expression is easily traceable by fluorescent microscopy and its nuclear localization allows for a specific blockage of TEAD without affecting cytoplasmic functions of YAP1 and TAZ.

Transduction of cells with a plasmid containing TEADi identified that this inhibitor is well expressed and localized to the nucleus (Fig. 1a) and can block basal TEAD-reporter activity as well as YAP1- and TAZ-induced TEAD activity in cells (Fig. 1b). Co-immunoprecipitation experiments showed that TEADi reduces the interaction of YAP1 and TAZ with TEAD, but it does not alter the interaction of these proteins with LATS1, a member of the Hippo pathway (Fig. 1c). Our results indicate that TEADi is a valid genetically encoded YAP1/TAZ-TEAD interaction inhibitor to study TEAD-dependent transcription and biological effects.

**TEAD activity maintains keratinocytes in a progenitor state.** To study the effects of inhibiting TEAD-dependent transcription in keratinocytes we took advantage of immortalized N/TERT2G keratinocytes that show similar epidermal differentiation in 2D culture and 3D organotypic skin models to human primary keratinocytes[24,25]. N/TERT2G keratinocytes transduced with adenoviruses expressing TEADi (ad-TEADi) showed a significant decrease in proliferation (Fig. 1d) accompanied by a marked increase in the mRNA expression of the differentiation markers keratin 1 (KRT1) and transglutaminase 3 (TGM3), and a reduction of the basal/progenitor markers keratin 5 (KRT5), compared with cells transduced with adenoviruses expressing GFP (ad-GFP) (Fig. 1e). We also observed a significant reduction in the mRNA expression of the YAP1/TAZ reporter genes CYR61 and CTGF (also known as CCN1 and CCN2 respectively, Fig. 1e) and reduced CYR61 protein expression (Supplementary Fig. 1a), indicating that TEADi blocks the activity of YAP1/TAZ-TEAD. TEADi expression also lead to a decrease in the protein expression of the basal markers p63 and KRT5, and an increase in the protein levels of the differentiation markers keratin 10 (KRT10) and involucrin (IVL) (Fig. 1f and Supplementary Fig. 1b, c), as well as a decrease in proliferation as measured by EdU-DNA incorporation and PCNA staining (Supplementary Fig. 1d, e). Blockage of TEAD by TEADi did not result in changes in the expression levels of YAP1 or TAZ (Fig. 1f) or nuclear localization of YAP1 (Supplementary Fig. 1f). Endogenous TEAD protein coimmunoprecipitation experiments in keratinocytes showed that TEADi reduces the interaction of YAP1 and TAZ with TEAD transcription factors (Supplementary Fig. 1g).

We next generated stable N/TERT2G keratinocytes expressing tetracycline-inducible TEADi or GFP as a control by lentiviral transduction and subjected these cells to 3D differentiation in cell culture inserts, which recapitulate the differentiation stages of skin in an in vitro system[25]. When N/TERT2G keratinocytes were induced to express TEADi, cells showed signs of early differentiation, resulting in thinner epidermal cultures with an aberrant and increased differentiation pattern (Fig. 1g). Immuno-fluorescence (IF) staining showed reduced expression of the basal marker KRT5 as well as reduced levels of the proliferation marker PCNA in organotypic cultures expressing TEADi (Fig. 1h and Supplementary Fig. 1h). Interestingly, cultures showed increase staining of the differentiation marker KRT10, which also labeled cells in the basal/progenitor layer (Fig. 1h and Supplementary Fig. 1i), indicating early activation of differentiation programs. However, cells retained the ability to form a layered epidermis in the absence of TEAD activity (Fig. 1g). Our results indicate that

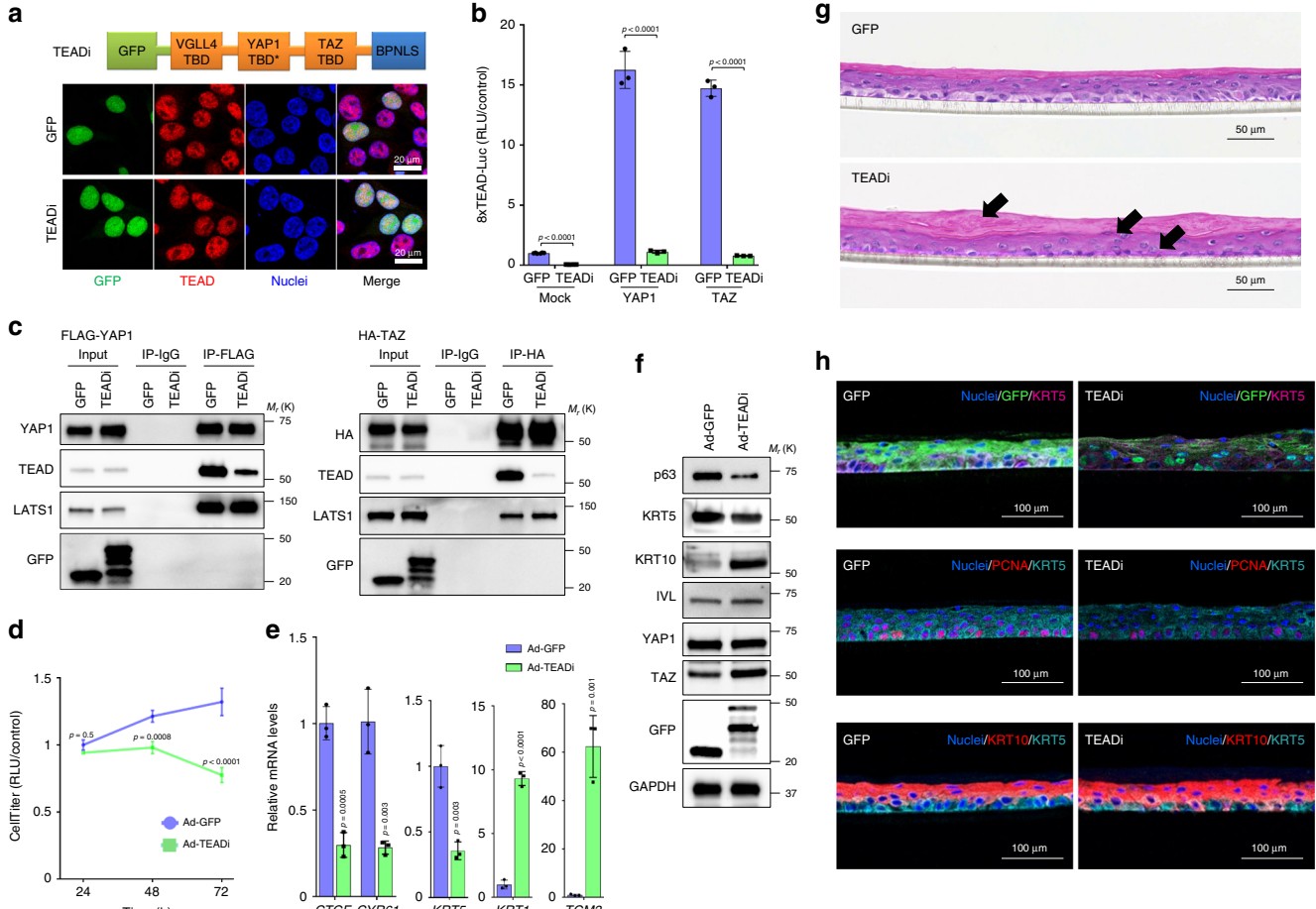

**Fig. 1 An inhibitor of the transcriptional activity of TEAD triggers rapid keratinocyte cell cycle arrest and differentiation. a** Schematic representation of TEADi and IF showing its nuclear localization in HEK293 cells. **b** Transcriptional activity of TEAD measured by luciferase assay with a reporter containing tandem TEAD-binding sites in HEK293 cells. **c** Coimmunoprecipitation experiments in HEK293 cells to show TEADi reduces TEAD interaction with YAP1 and TAZ. **d** Luminescence cell proliferation assay in N/TERT2G cells transduced with adenoviruses expressing GFP (Ad-GFP, control) or TEADi (Ad-TEADi). **e** qRT–PCR analysis of expression of indicated markers in N/TERT2G keratinocytes transduced with GFP (Ad-GFP, control) or TEADi (Ad-TEADi) for 48 h. Values are indicated in fold over GFP control. **f** Western blot analysis of expression of basal and differentiation markers in N/TERT2G cells transduced with GFP (Ad-GFP, control) or TEADi (Ad-TEADi) for 48 h. **g** Epidermis reconstruction assay with N/TERT2G keratinocytes stably expressing a tetracycline-inducible GFP (control) or TEADi. Arrows indicate aberrant differentiation. **h** IF staining showing the expression of proliferation and differentiation markers in the organotypic epidermis. In **b**, mock $n = 6$, YAP1 $n = 3$, TAZ $= 3$ biological replicates each; in **d** and **e** $n = 3$ biological replicates. Mean ± SD is shown; in **b** and **e** two-tailed unpaired $t$ test used and in **d** two-way ANOVA with Sidak's multiple comparison test. Source data are provided as a Source Data file.

TEAD interaction with YAP1/TAZ is necessary to maintain keratinocytes in a basal, undifferentiated state.

**TEAD transcriptional networks in keratinocytes.** RNA sequencing (RNA-seq) was performed to characterize the transcriptional effects of TEAD inhibition in N/TERT2G keratinocytes transduced with ad-TEADi or control ad-GFP after 12, 24 and 48 h. Differentially regulated genes were considered as having an absolute fold change ($|FC| \geq 1.5$) and a false discovery rate (FDR) adjusted $q$-value ($q$) < 0.05. We could not observe any significant differentially-regulated genes at 12 h, probably due to a low expression of TEADi at this time point (Fig. 2a), however the expression of several genes was affected by TEAD inhibition at 24 and 48 h, with a significant overlap of differentially regulated genes (Fig. 2b and Supplementary Data 1). TEADi early response transcripts present in both the 24 and 48 h datasets included several genes involved in epithelial differentiation, including

*NOTCH3*, *FOXN1*, *ZNF750*, *IVL*, *CNFN*, and *DSC2*, as well as the YAP1/TAZ response genes *CYR61* and *CTGF*.

Comparing the differentially regulated genes following TEADi expression at 48 h in keratinocytes with those found to be regulated by YAP1 in other studies[13,26], identified a core of well-established genes regulated by YAP1 and TAZ, which included *AXL*, *CTGF*, *CYR61*, and *FST* shared among the three datasets (Fig. 2c). To further compare the transcriptional consequences of TEAD blockage with those triggered by loss of YAP1 and TAZ, we performed RNA-seq in N/TERT2G keratinocytes with YAP1 and TAZ knockdown by pooled-small interfering RNA (siYAP1/TAZ, Supplementary Fig. 2a and Supplementary Data 2). As expected, siYAP1/TAZ reduced keratinocyte proliferation (Supplementary Fig. 2b) and lead to gene expression changes that produced a larger amount of significant differentially regulated genes than TEADi (Fig. 2d). However, significant overlap was observed between differentially regulated genes in both conditions (Fig. 2d), confirming that common gene networks are

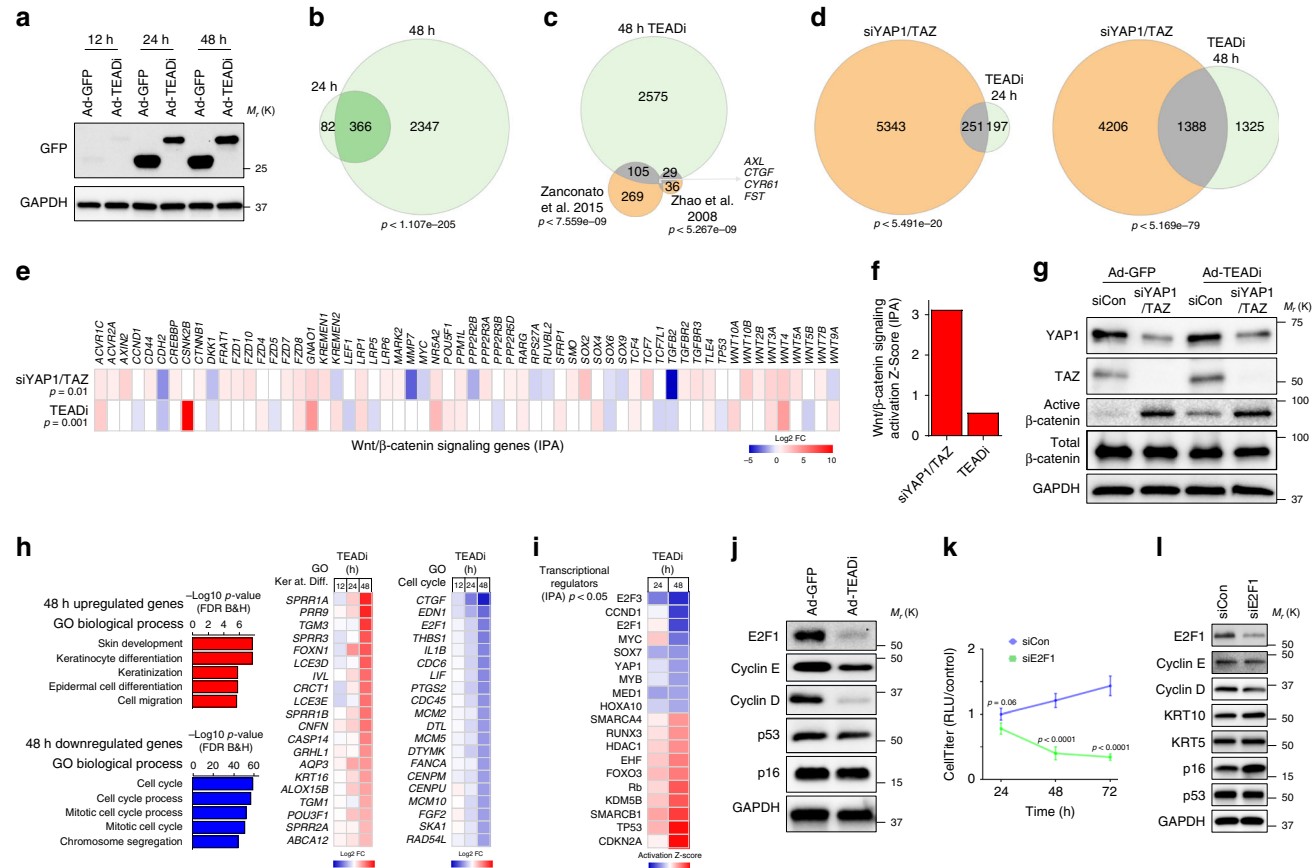

**Fig. 2 TEAD transcriptional networks in keratinocytes regulate cell cycle entry and differentiation programs. a** Western blot analysis of expression of GFP (Ad-GFP, control) and TEADi (Ad-TEADi) in N/TERT2G cells. **b** Venn diagram showing the overlap between differentially regulated genes in TEADi vs GFP ($q < 0.05$, $|FC| \geq 1.5$) in N/TERT2G keratinocytes at 24 and 48 h following transduction. **c, d** Venn diagrams showing the overlap between differentially regulated genes ($q < 0.05$, $|FC| \geq 1.5$) by TEADi expression in N/TERT2G keratinocytes and published YAP1 signatures[13,26] (**c**) or siYAP1/TAZ (**d**). **e** Graph indicating the fold change (Log2 FC) of genes related to canonical Wnt/β-catenin signaling (IPA) in the siYAP1/TAZ and TEADi 48 h datasets. Only differentially regulated genes are shown. White color indicates the gene is not differentially regulated in that dataset. The IPA significance p-value ($p$) of overlap for the Wnt/β-catenin canonical pathway is indicated for each dataset. **f** IPA activation Z-score for canonical Wnt/β-catenin signaling in the indicated datasets. **g** Western blot analysis of expression of the indicated markers in N/TERT2G cells. Active β-Catenin refers to non-phospho β-Catenin at Ser45. **h** Graph indicating the GO biological process terms enriched in upregulated ($q < 0.05$, FC $\geq 1.5$) and downregulated ($q < 0.05$, FC $\leq -1.5$) genes by TEADi expression in N/TERT2G keratinocytes at 48 h and fold change (Log2 FC) of genes related to GO keratinocyte differentiation and cell cycle at corresponding times compared with GFP expressing cells. **i** IPA functional analysis of activation state of upstream transcriptional regulators for genes in **b**. **j** Western blot analysis of the expression of cell cycle regulators in N/TERT2G cells transduced with GFP (Ad-GFP, control) or TEADi (Ad-TEADi) for 48 h. **k** Luminescence cell proliferation assay in N/TERT2G cells transduced with pooled siRNAs for E2F1. **l** Western blot analysis of the expression of cell cycle regulators and differentiation markers in N/TERT2G cells transduced with pooled siRNAs for E2F1 for 48 h. In **k** $n = 3$ biological replicates. Mean ± SD is shown; two-way ANOVA with Sidak's multiple comparison test. In **b**, **c** and **d** $p$ indicates the p-value of the overlap in Venn diagrams, Fisher's exact test. Source data are provided as a Source Data file.

present in TEADi and siYAP1/TAZ datasets. Additional non-overlapping effects between TEADi and siYAP1/TAZ can be due to numerous reasons, including additional compensatory mechanisms that are differentially activated in conditions in which YAP1 and TAZ are absent, conditions that are not the same when TEAD binding is blocked and YAP1 and TAZ remain present to interact with other factors; timing of a protein inhibitor (TEADi) versus a gradual decrease in protein expression (siRNA); and the fact that siRNA does not completely depletes YAP1 protein expression (Supplementary Fig. 2a).

To better illustrate the differences between TEADi and YAP1 and TAZ knockdown that can lead to non-overlapping effects we focused on the Wnt/β-catenin pathway. Regulation of the activation of β-catenin has been shown to be independent of TEAD transcription factors and dependent on the direct interaction of YAP1 and TAZ with components of the Wnt

signaling pathway[18,19]. Analysis of canonical pathways affected by siYAP1/TAZ and TEADi in Ingenuity Pathway Analysis (IPA) indicated that, although under both conditions Wnt/β-catenin signaling gene networks are differentially regulated (Fig. 2e), siYAP1/TAZ leads to a clear increase in the activity of the pathway when compared with TEADi (Fig. 2f). Indeed, several downstream targets of Wnt signaling were activated by siYAP1/TAZ and not TEADi, including *AXIN2* and *CD44* (Fig. 2e). By utilizing an antibody that recognizes non-phospho active β-Catenin (Ser45)[27], we confirmed that siYAP1/TAZ results in a significant increase in the amount of active β-catenin in keratinocytes, while TEADi results in minimal alterations in the levels of active protein (Fig. 2g and Supplementary Fig. 2c). Our results indicate that YAP1/TAZ modulate Wnt signaling through TEAD-dependent and independent events and confirm that TEADi can discern YAP1/TAZ-TEAD specific effects.

We next focused on the TEAD-dependent gene networks in keratinocytes. Gene ontology (GO) analysis of the differentially regulated genes in the TEADi dataset at 48 h identified that TEAD inhibition induces a rapid activation of differentiation and skin development gene networks and a downregulation of genes involved in cell cycle progression and processes (Fig. 2h). TEAD blockage resulted in a rapid increase in the expression levels of differentiation markers including *TGM1* and *3* and involucrin (*IVL*) and reduced the expression levels of several cell cycle regulators, including *E2F1* (Fig. 2h). Analysis of gene sets common to TEADi and siYAP1/TAZ, which would constitute the core of YAP1/TAZ-TEAD transcriptional activity, showed similar GO terms related to cell cycle and differentiation (Supplementary Fig. 2d). The overlap in the effects of TEADi and siYAP1/TAZ was validated even when we selected the top genes modified by TEADi and siYAP1/TAZ by applying more stringent threshold levels ($|FC| \geq 2$ and $q < 0.01$, Supplementary Fig. 2e), and indicate that the most significant processes differentially regulated under both conditions are related to adhesion and differentiation (Supplementary Fig. 2e).

Analysis of upstream regulators by IPA in the TEADi 48 h dataset identified that transcriptional networks related to E2F, cyclins, YAP1, and MYC were downregulated, while networks related to cell cycle inhibitors were upregulated (Fig. 2i). Indeed, TEAD inhibition results in a reduction of E2F1 and cyclin D levels (Fig. 2j). However, we were not able to detect differences in p53, p16, apoptosis markers or phospho-Rb levels, (Fig. 2j and Supplementary Fig. 2f, g), indicating that the inhibition of cell cycle could be a direct result of the transcriptional effect of TEAD on the expression of E2F cell cycle regulators. Supporting a direct role of TEAD in controlling E2F transcription, TEADi expression resulted in a reduction in the mRNA levels of several members of the E2F family, including *E2F1*, *2*, *5* and *8* (Supplementary Fig. 2h), and in a decrease in luciferase signal in a reporter containing the human *E2F1* promoter (Supplementary Fig. 2i, j). Knockdown of E2F1 (siE2F1) is sufficient to reduce cell growth in keratinocytes (Fig. 2k, l), indicating a critical role for E2F1 regulating cell cycle entry. Interestingly, while reduced levels of E2F1 halted keratinocyte proliferation and reduced the levels of cyclin D expression, it did not result in increased expression of the differentiation marker KRT10 or reduced expression of the basal marker KRT5 (Fig. 2l).

Overall, our results indicate that TEADi can be used to dissect YAP1/TAZ-TEAD specific gene networks and that TEAD controls cell cycle progression in keratinocytes by regulating the levels of *E2F1*.

**TEAD and KLF4 limit each other activity**. YAP1/TAZ-TEAD complexes can modulate transcription at several levels, including direct promoter regulation, enhancer association, or binding with chromatin regulating proteins[26,28,29], indicating that TEAD can control epithelial homeostasis indirectly by regulating the expression and chromatin accessibility of other transcription factors affecting central cell renewal and differentiation gene networks.

To dissect global transcriptional networks affected by TEAD-blockage we performed a transcription factor binding site enrichment analysis in the TEADi upregulated and downregulated gene sets (Fig. 3a and Supplementary Fig. 3a–d). We found that only about 19% of the downregulated genes and 25% of the upregulated genes at 48 h present predicted TEAD-binding sites in their promoter regions (Supplementary Fig. 3a, b). Over-represented conserved transcription factor binding sites in the downregulated gene set included AP1, TEAD1, Myc and E2F1 (Fig. 3a and Supplementary Fig. 3c), supporting the role of TEAD

in facilitating transcription from factors important for keratinocyte self-renewal and proliferation. Interestingly, binding sites in upregulated genes in keratinocytes showed a clear enrichment for KLF4 (Fig. 3a and Supplementary Fig. 3d). While KLF4 is a well-known stem cell factor in embryonic and induced pluripotent stem cells, in keratinocytes is central to the specification of differentiated cells[30,31], suggesting that YAP1/TAZ-TEAD could regulate differentiation by limiting the activity of KLF4.

Inhibition of TEAD transcriptional activity by TEADi in keratinocytes resulted in increased nuclear KLF4 expression at 48 h (Fig. 3b) and KLF4 was necessary for the activation of differentiation following TEAD inhibition as measured by KRT10 expression (Fig. 3c). Interestingly, knockdown of KLF4 resulted in increased expression of the YAP1/TAZ-TEAD target CYR61 (Fig. 3c), indicating the possibility of a dual regulation in which TEAD and KLF4 limit each other activity. Supporting this, KLF4 knockdown resulted in increased association between YAP1 and TEAD (Fig. 3d). In contrast, overexpression of KLF4 in keratinocytes lead to an increase differentiation and reduced levels of CYR61 (Fig. 3e). Consistent with KLF4 regulating the transcriptional activity of TEAD, knockdown of KLF4 in HEK293 cells results in a significant increase in TEAD-reporter activity (Fig. 3f).

To confirm the mutual regulation of transcriptional networks under TEAD and KLF4, we performed RNA-seq in N/TERT2G keratinocytes with siKLF4 (Supplementary Data 3). KLF4 knockdown resulted in the upregulation of genes related to cell cycle and division and downregulation of transcripts that participate in epidermis development and biological adhesion (Fig. 3g). IPA analysis of upstream networks affected by siKLF4 indicated an activation of transcriptional targets of MITF, E2F factors and YAP1 (Fig. 3h), confirming that reduction in KLF4 levels leads to an increase transcriptional activity downstream of YAP1. Indeed, differentially regulated genes in siKLF4 showed a significant overlap with genes differentially regulated by TEADi at 48 h and by siYAP1/TAZ (Fig. 3i), further confirming that the transcriptional networks downstream from KLF4 and TEAD-YAP1/TAZ are closely intertwined. Since KLF4 is primarily involved in the specification of differentiated cells, we analyzed the expression of differentiation markers and found that TEADi and siYAP1/TAZ consistently lead to upregulation of key differentiation genes, including *IVL*, *FOXN1*, *KRT1,* and *KRT10*, while siKLF4 results in downregulation of these transcripts (Fig. 3j). It is worth noting that siYAP1/TAZ leads to an increase in KLF4 mRNA and protein levels and activation of differentiation (Fig. 3j, k), which recapitulates the effect of TEADi but is not further increased by the expression of the TEAD inhibitor (Fig. 3k), confirming that the effects observed by TEADi are indeed mediated by YAP1/TAZ activity.

Our results indicate that the balance of transcriptional activity between KLF4 and YAP1/TAZ-TEAD is critical for the activation of differentiation gene networks, suggesting a mechanism could be in place by which they can directly modulate each other activity. It has been shown that YAP1/TAZ cooperate with KLF4 to promote the differentiation of mouse intestinal cells into goblet cells by direct binding of YAP1/TAZ to KLF4[32]. Co-immunoprecipitation experiments demonstrated that KLF4 interacts with TEAD, YAP1 and TAZ in keratinocytes and that this interaction is increased by TEADi (Fig. 4a). Our results indicated two scenarios for TEAD and KLF4: one in which YAP1/TAZ and TEAD binding to KLF4 can affect the activity of this differentiation factor, an another one in which KLF4 could bind to YAP1/TAZ-TEAD complexes to limit their transcriptional activity. To demonstrate whether KLF4-YAP1/TAZ interaction has a functional consequence on KLF4 activity, we constructed GAL4-DNA binding domain fusion proteins (indicated as GAL4)

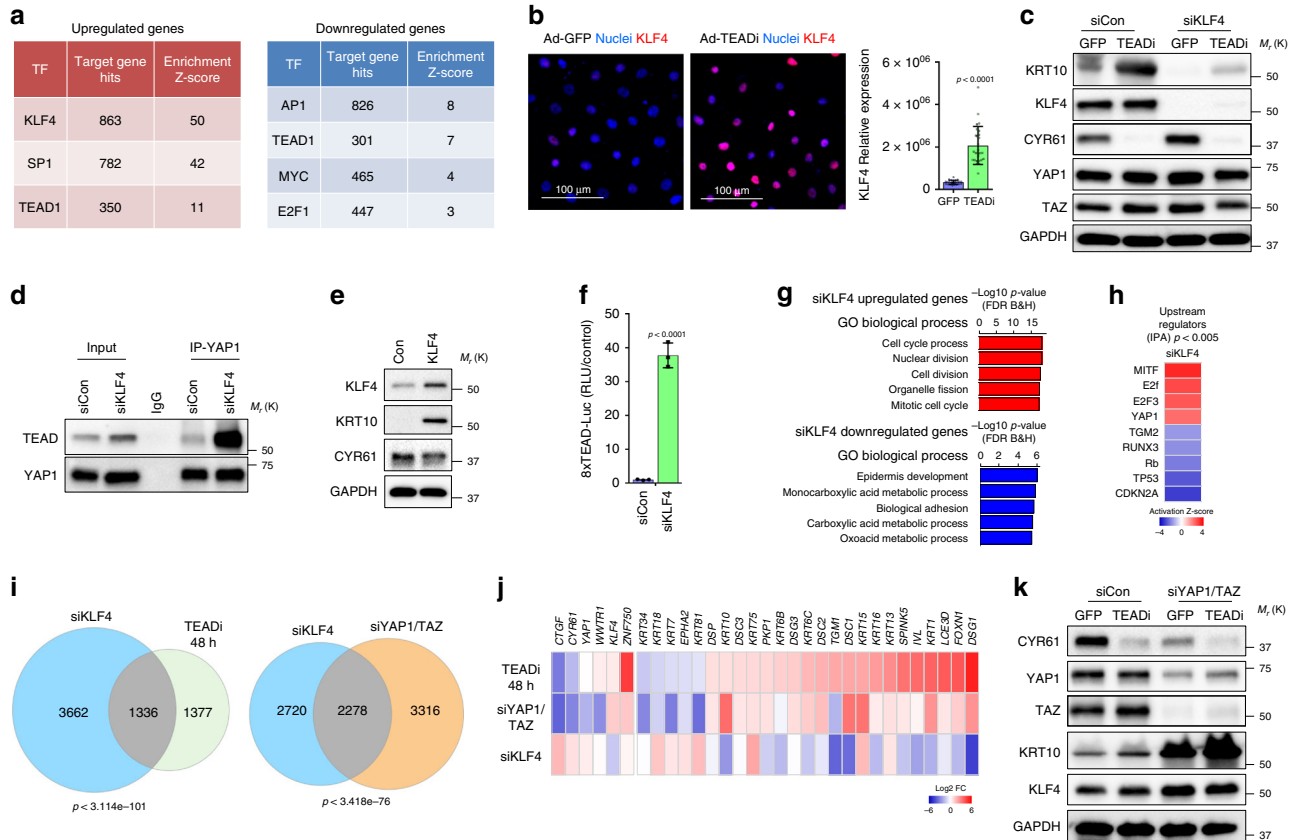

**Fig. 3 Regulation of keratinocyte differentiation by TEAD and KLF4. a** Selected factors found in the transcription factor binding site enrichment analysis of TEADi upregulated and downregulated gene sets. **b** IF staining and quantification showing the expression of KLF4 in N/TERT2G cells following transduction with GFP (Ad-GFP, control) or TEADi (Ad-TEADi) for 48 h. **c** Western blot analysis of expression of indicated markers in N/TERT2G cells transduced with pooled siRNAs for KLF4 for 24 h and GFP (Ad-GFP, control) or TEADi (Ad-TEADi) for additional 48 h. **d** Coimmunoprecipitation experiments in N/TERT2G keratinocytes transduced with pooled siRNAs for KLF4 for 48 h. **e** Western blot analysis in N/TERT2G keratinocytes transduced with lentiviruses expressing GFP or KLF4 for 48 h. **f** Transcriptional activity of TEAD measured by luciferase assay with a reporter containing tandem TEAD-binding sites in HEK293 cells transduced with pooled siRNAs for KLF4 for 48 h. **g** Graph indicating the GO biological process terms enriched in upregulated ($q < 0.05$, FC $\geq 1.5$) and downregulated ($q < 0.05$, FC $\leq -1.5$) genes by KLF4 knockdown (siKLF4) in N/TERT2G keratinocytes. **h** Selected upstream transcriptional regulators for genes in **g** generated using IPA. **i** Venn diagrams showing the gene number overlap of differentially regulated genes ($q < 0.05$, |FC| $\geq 1.5$) between the indicated datasets in N/TERT2G keratinocytes. **j** Graph indicating the fold change (Log2 FC) of genes related to keratinocyte differentiation in the indicated datasets. Only differentiation genes differentially regulated in the TEADi 48 h dataset are shown. **k** Western blot analysis in N/TERT2G keratinocytes treated with or without siYAP1/TAZ and then transduced with adenoviruses expressing GFP or TEADi. In **b** n = 27 fields from 3 biological replicates, (**f**) n = 3 biological replicates. Mean ± SD is shown; two-tailed unpaired t test. In (**i**) p indicates the p-value of the overlap in Venn diagrams, Fisher's exact test. Source data are provided as a Source Data file.

with the 329 N-terminal amino acids of KLF4, which harbor the activation and repression domains but excludes the zinc finger domain[33], and two fragments containing only the KLF4 activation domain (1–156) or only the repressor domain (159–329) (Fig. 4b). Overexpression of wild type YAP1 or a YAP1 S94/F95 mutant that does not bind to TEAD[34], resulted in a marginal potentiation of GAL4-KLF4 1–329 and GAL4-KLF4 1–156 activity, while the construct containing the repressor domain (GAL4-KLF4 159–329) did not induce transcription (Fig. 4c). This marginal increase in activity, coupled to the fact that keratinocytes can activate KLF4 and differentiation in the absence of YAP1/TAZ (Fig. 3j, k), indicates that YAP1/TAZ interaction with KLF4 might not have functional consequences during keratinocyte differentiation. On the other hand, KLF4 was able to reduce YAP1-induced expression of CYR61 and activation of TEAD transcription, particularly GAL4-KLF4 1–329 and GAL4-KLF4 1–156 (Fig. 4d, e). KLF4 lacking the DNA-binding domain was still able to bind to TEAD, particularly the activation

domain of KLF4 (Fig. 4f). Collectively, these results expose a scenario where KLF4 binds YAP1/TAZ-TEAD complexes to limit TEAD transcription.

**TEADi disrupts epithelial homeostasis in the mouse skin.** To define the role of YAP1/TAZ-TEAD transcription in specific tissues in vivo we developed a tetracycline-inducible TEADi transgenic mouse. We targeted expression of TEADi to the epidermis and its stem cells by breeding our TEADi transgenic line with mice expressing the reverse tetracycline-inducible transactivator (rtTA2) under the control of the cytokeratin 5 promoter (KRT5rtTA)[35,36] (Fig. 5a). Mice treated with doxycycline chow showed expression of TEADi in the nuclei of basal cells in the interfollicular epidermis and hair follicles (Fig. 5b). After 10–20 days of induction, mice showed changes in hair color, hair loss, skin ulcer formation, and general discomfort, indicating disruption of skin homeostasis by YAP1/TAZ-TEAD blockage.

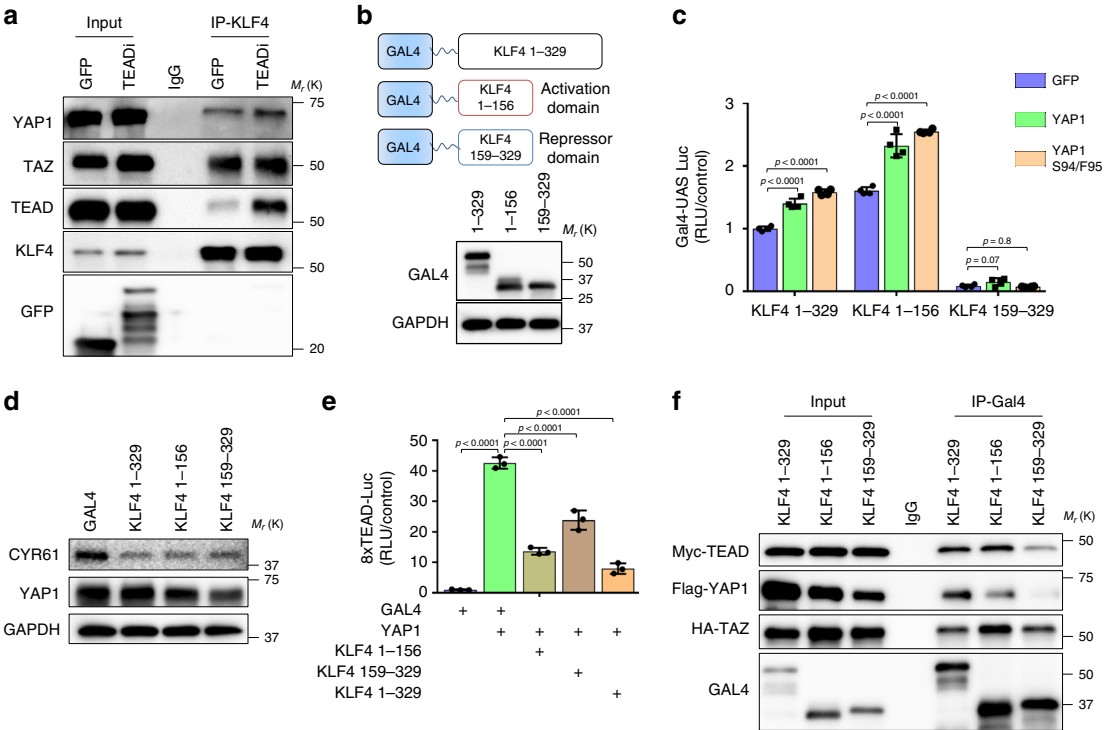

**Fig. 4 KLF4 interacts with the YAP1/TAZ-TEAD complex. a** Coimmunoprecipitation experiments in N/TERT2G cells transduced with TEADi or GFP. **b** Schematic representation of GAL4-KLF4 fusion proteins and western blot showing their expression in HEK293 cells. **c** Transcriptional activity of GAL4 DNA-binding domain fusion proteins measured by luciferase assay by UAS-Luc in HEK293 cells. **d** Western blot analysis of the expression the indicated markers in HEK293 cells transduced YAP1 in combination with the indicated GAL4 fusion proteins for 48 h. **e** Transcriptional activity of TEAD in HEK293 cells transduced with YAP1 and GAL4-KLF4 fusion proteins or GAL4 control. **f** Coimmunoprecipitation experiments in HEK293 cells transduced with the indicated GAL4-KLF4 proteins and Myc-tagged TEAD, HA-tagged TAZ and FLAG-tagged YAP1. In **c** $n = 4$ biological replicates; in **e** $n = 3$ biological replicates. Mean ± SD is shown; one-way ANOVA with Dunnett's multiple comparisons test. Source data are provided as a Source Data file.

Histological analysis showed skin thinning 4 days following TEADi induction, and ulcer formation and areas of reactive thickened epidermis surrounding the wounds in mice after 12 days (Fig. 5b). Quantification of basal cell proliferation by PCNA staining demonstrated that TEADi expression leads to a significant reduction in basal keratinocyte cell proliferation as early as 4 days, and this reduction is sustained after 12 days (Fig. 5c). One possible explanation for the observed phenotype is that reduced proliferation of basal keratinocytes leads to stem/progenitor cell depletion in the epidermis, disrupting skin homeostasis and facilitating wound formation. Indeed, we observed low levels of the basal marker KRT5 at 4 and 12 days following TEADi expression (Fig. 5d), accompanied by an increase in expression of the differentiation markers KRT10 and loricrin (Fig. 5d and Supplementary Fig. 4a). Furthermore, the levels of the basal and stem cell marker p63[37] were rapidly reduced by TEADi expression in the epidermis (Fig. 5e) and isolated keratinocytes from mice after 4 days of TEADi expression showed reduced clonogenic capacity (Supplementary Fig. 4b), indicating stem/progenitor cell depletion. Aligned with our in vitro results, TEADi also lead to a marked increase in KLF4 expression as early as 4 days after induction and thickened epidermal areas in 12-days mice also showed increased KLF4 expression (Fig. 5f).

Remarkably, despite the observed stem/progenitor cell depletion, mice that developed ulcers after 10–20 days of TEADi induction showed thickened epidermis in several areas surrounding wounds (Fig. 5a and Fig. 6a). These thickened epidermal areas were mostly composed by cells positive for KRT10 and loricrin and negative for the basal markers p63 and KRT5 (Fig. 6a and Supplementary Fig. 4a), indicating that they are differentiated

cells. One possible explanation for these thickened and differentiated epidermal areas is that cells transitioning into differentiation with low expression of TEADi can still proliferate. Indeed, we were able to detect proliferating cells in the epidermis of mice at 12 days that present low levels of KRT5 staining (Figs. 5c and 6b). In addition, proliferating areas at 12 days were negative for TEADi expression (Fig. 6c) and we observed a reduction over time of cells expressing detectable levels of TEADi (Fig. 6d), probably caused by the fact that TEADi positive cells rapidly differentiate, leading to a termination of KRT5rtTA expression and, consequently, TEADi downregulation. Of interest, labeling of lymphoid cells and macrophages in the skin with CD45 indicated an increase in inflammatory cell infiltration by 4 days that is further increased by 12 days following TEADi expression (Fig. 6e). Inflammatory cell infiltration is commonly caused by the disruption of epithelial integrity and could be a plausible trigger for the observed keratinocyte activation and thickened epidermal areas at late time points.

Altogether, our results indicate that blockage of TEAD activity in the basal compartment of the skin leads to reduced proliferation and increased differentiation of basal cells, resulting in the depletion of KRT5[+]/p63[+] progenitor/stem cells and disruption of epithelial integrity.

## Discussion

The precise study of the transcriptional networks regulated downstream of YAP1 and TAZ in somatic stem cells and cancer, particularly in the skin, is hindered by numerous factors. First, knockout of YAP1 is sufficient to disrupt stem cell function and skin homeostasis in developing mice[7], whereas both YAP1 and

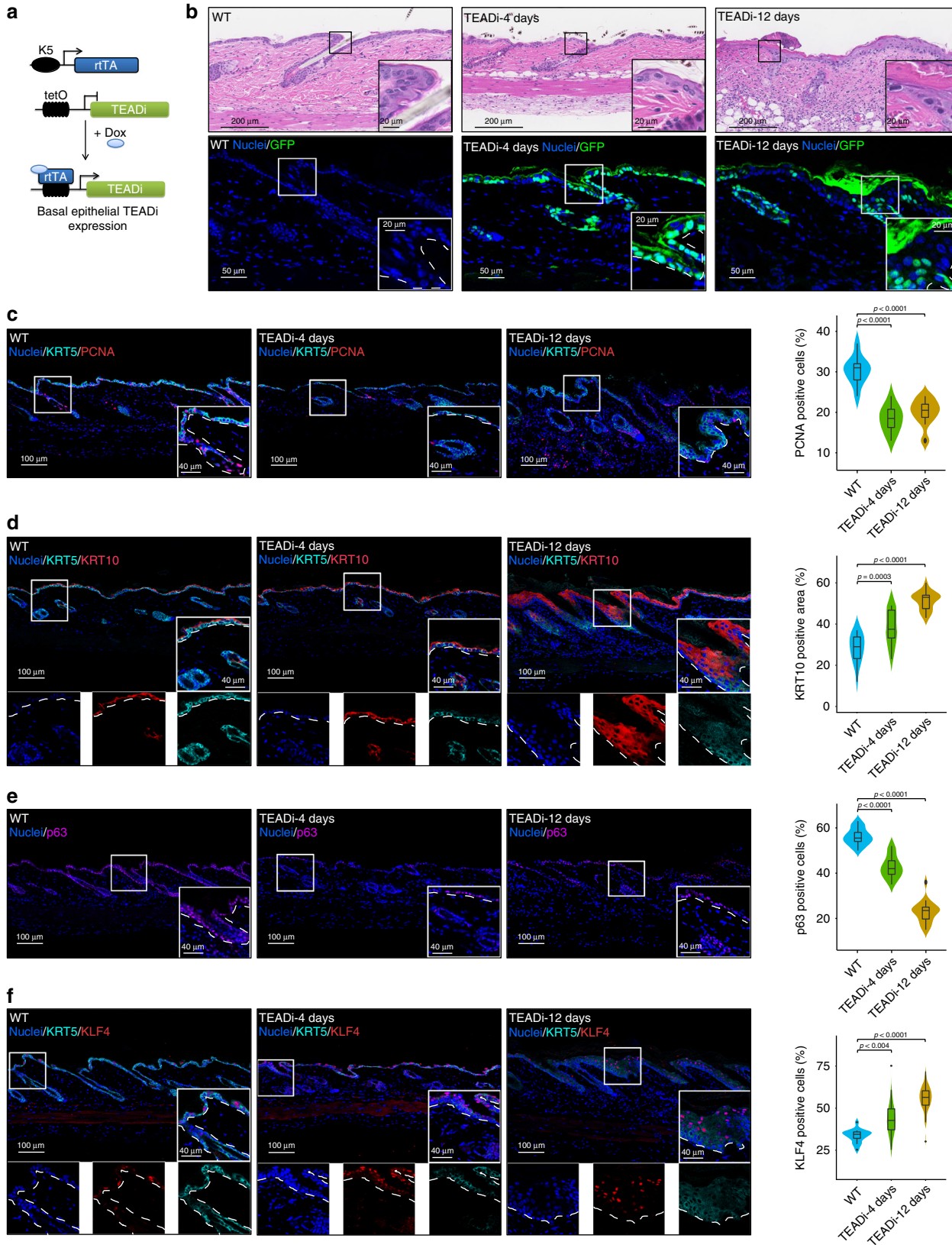

TAZ have to be downregulated to observe an effect in adult skin, BCC and SCC[8,9,11]. Consequently, the generation of double knockout mice is required to characterize the effects of YAP1/TAZ in adult tissues. Second, YAP1/TAZ interact with numerous effectors that regulate not only gene transcription but also other processes[16–19], indicating that YAP1/TAZ knockout results in countless transcriptional, structural and signaling effects, which are usually cumbersome to dissect. Finally, it has been suggested

**Fig. 5 TEAD blockage in the mouse skin induces keratinocyte differentiation and progenitor cell depletion. a** Schematic representation of the animal model used to target the inducible expression of the TEADi to the basal epidermal stem cell compartment. **b** Histological analysis of wild type (WT) mice and mice induced to express TEADi for 4 and 12 days and IF showing nuclear expression of TEADi. **c–f** Representative IF pictures of the indicated markers and corresponding expression quantification from the skin of WT or TEADi mice induced for 4 or 12 days. Inserts show magnification of highlighted area and location of the basal membrane is indicated with a white dotted line. In **d** and **f** the different fluorescent channels of the insert magnification are presented individually bellow the picture for better clarity. In **c** WT $n = 10$, TEADi-4d $n = 11$, TEADi-12d $n = 16$; (**d**) WT $n = 14$, TEADi-4d $n = 18$, TEADi-12d $n = 19$; (**e**) WT $n = 14$, TEADi-4d $n = 15$, TEADi-12d $n = 16$; and **f** WT $n = 14$, TEADi-4d $n = 18$, TEADi-12d $n = 19$ fields from 3 different mice in each condition. Violin plots shows density of data and box plot shows median, 25 and 75% quartile, 95% confidence interval and dots indicating potential outliers; two-way ANOVA with Dunnett's multiple comparison test. Source data are provided as a Source Data file.

that YAP1 knockout mice could still express shorter isoforms of YAP1 protein, accounting for phenotypic differences in mouse studies[32]. To overcome these roadblocks, here we employ the use of TEADi, a genetically encoded inhibitor of the interaction of YAP1/TAZ with one of their main downstream effectors, TEAD transcription factors, to dissect the specific regulatory gene expression networks downstream of YAP1/TAZ-TEAD.

We demonstrate that TEAD activity regulates epithelial cell homeostasis at two independent levels (Fig. 7a): by controlling the expression of key factors necessary for cell cycle entry and proliferation, and via a regulatory loop with master regulators of commitment to differentiation. TEAD regulates the transcription of the *E2F1* gene and *E2F1* knockdown recapitulates the cell cycle blockage produced by TEAD inhibition, indicating that E2F1 is a central regulator for proliferation in keratinocytes. However, *E2F1* knockdown is not sufficient to trigger the rapid activation of differentiation programs observed by TEADi expression.

Blockage of YAP1/TAZ binding to TEAD resulted in activation of KLF4 transcriptional networks, a central factor necessary for differentiation commitment[30,31], and we demonstrate that KLF4 and YAP1/TAZ-TEAD transcriptional networks are closely intertwined. Our results suggest that TEAD and KLF4 transcription factors are at the crossroad of a regulatory loop that determines the differentiation state of keratinocytes. KLF4 is downregulated during skin cancer progression and KLF4 knockout mice show increase cancer incidence[30], pointing towards a role of KLF4 regulating both activation of differentiation programs and direct repression of pathways involved in epithelial stem cell maintenance, including YAP1/TAZ-TEAD. Furthermore, given that YAP1/TAZ-TEAD transcriptional networks are the downstream effector of numerous chemical and biomechanical signals, TEAD could serve as the link between KLF4 activity and microenvironmental chemical and mechanical cues. Although we demonstrate that KLF4 can directly bind and regulate the activation of YAP1/TAZ-TEAD complexes, the precise mechanism by which KLF4 acts and by which TEAD regulates KLF4 expression and activity requires further investigation.

Our results also highlight the essential role of TEAD transcriptional regulation to maintain epithelial homeostasis in vivo. TEAD activity is necessary to keep basal progenitor cell proliferation and renewal, and inhibition of TEAD results in rapid differentiation and barrier disruption, leading to immune cell infiltration. Interestingly, despite the reduction in basal cell proliferation, both human and mouse keratinocytes are able to form a layered epidermis in the presence of TEADi, although with a significant increase in differentiated cells. This phenotype could be the result of heterogeneous levels of TEAD inhibition across basal cells or the expansion of transient amplifying cells that might not require TEAD activity to proliferate.

The peptides used to build TEADi were selected due to their proven specificity towards TEAD binding[20–22] and our results suggest that the main effects of TEADi are indeed mediated by YAP1/TAZ-TEAD activity. One limitation of our model is that despite the specificity of the TBDs they could cause additional nuclear effects that are not mediated by YAP1/TAZ, particularly the TBD of VGLL4. It has been shown in numerous studies that the main function of VGLL4 in cells is to block the binding of YAP1/TAZ to TEAD transcription factors[21,38–42], indicating that TBD-VGLL4 should not have any additional effects than to potentiate endogenous VGLL4-mediated TEAD inhibition. In addition, VGLL4 knockout mice have no reported phenotype on skin development[42], suggesting that blockage of VGLL4 in keratinocytes might not have any functional consequences. It is worth noting also that TEADi and YAP1/TAZ knockdown will cause additional non-overlapping effects due to the fact that both are different conditions (Fig. 7b): in TEADi expressing cells, YAP1 and TAZ remain present to interact with other factors; while in siYAP1/TAZ cells, structural interactions of YAP1 and TAZ, including interactions with the β-Catenin destruction complex, are altered, causing additional effects not present in TEADi cells.

Considering that the Hippo pathway constitutes one of the top signaling pathways altered in human cancer[12], disruption of YAP1/TAZ-TEAD complexes has become a main target to suppress oncogenic activity. TEADi could potentially be used to dissect the TEAD-dependent and independent roles of YAP1/TAZ signaling and aid in the discovery of improved targeting strategies for this pathway in cancer and other pathologies. In conclusion, the use of TEADi could become an additional invaluable resource for studying YAP1/TAZ-TEAD dependent transcription, with improved advantages that include rapid and simple inhibition of TEAD transcription and specific blockage of nuclear events mediated by both YAP1 and TAZ without affecting structural or cytoplasmic functions of these proteins.

## Methods

**DNA constructs**. TEADi was cloned using a gBlocks Gene Fragment (Integrated DNA Technologies) coding for the following amino acid sequence downstream of GFP into a pCEFL vector: SVDDHFAKALGDTWLQIGDPPVATNPKTANVPQT VPMRLRKLPDSFFKEPEGDPPVATNPKPSSWRKKILPESFFKEPGDPPVATKRT ADGSEFESPKKKRKVE. SVDDHFAKALGDTWLQI corresponds to the human VGLL4-TEAD binding domain (TBD VGLL4)[21], NPKTANVPQTVPMRLRKLPD SFFKEPE corresponds to the human YAP1-TEAD binding domain with a Pro98Glu change that increases binding to TEAD (TBD YAP1*)[20,21], NPKPSSWRKKILP ESFFKEP corresponds to the human TAZ-TEAD binding domain (TBD TAZ)[20], KRTADGSEFESPKKKRKVE corresponds to the bipartite nuclear-localization signal (BPNLS)[23], and GDPPVAT corresponds to a flexible linker sequence. pCEFL vector was a gift from Silvio Gutkind. TEADi tetracycline-inducible lentiviral construct (Lenti-TRE-TEADi) was made by cloning TEADi into pInducer20, which was a gift from Stephen Elledge (Addgene plasmid #44012)[43]. 8xTEAD-Luc (pGL3b 8xGTIIC-luciferase) was a gift from Stefano Piccolo (Addgene plasmid #34615). Human YAP1 with a C-terminal FLAG (pCEFL FLAG-YAP1) was a gift from Silvio Gutkind and has been reported previously[10]. HA-TAZ[26], pCMV-Flag-YAP1-S94/F95A and Myc-TEAD4 were a gift from Kunliang Guan (Addgene plasmids #32839, #33057 and #24638 respectively). For the E2F1 promoter reporter, a region of ~1.7 kb upstream of the transcription initiation site of the human *E2F1* gene was cloned by PCR from human genomic DNA (Bioline) with the following primers: forward 5′GCTGGTA CACCAGTTTGCTTT3′, reverse 5′TTTTGCCGCGAAAGAGCC3′; this region was then cloned into the pGL4.21(luc2P/Puro) vector (Promega). GAL4-KLF4 constructs were cloned by PCR of corresponding human KLF4 amino acids downstream of the DNA-binding domain of GAL4 into a pCEFL vector. pCEFL GAL4dbd was a gift

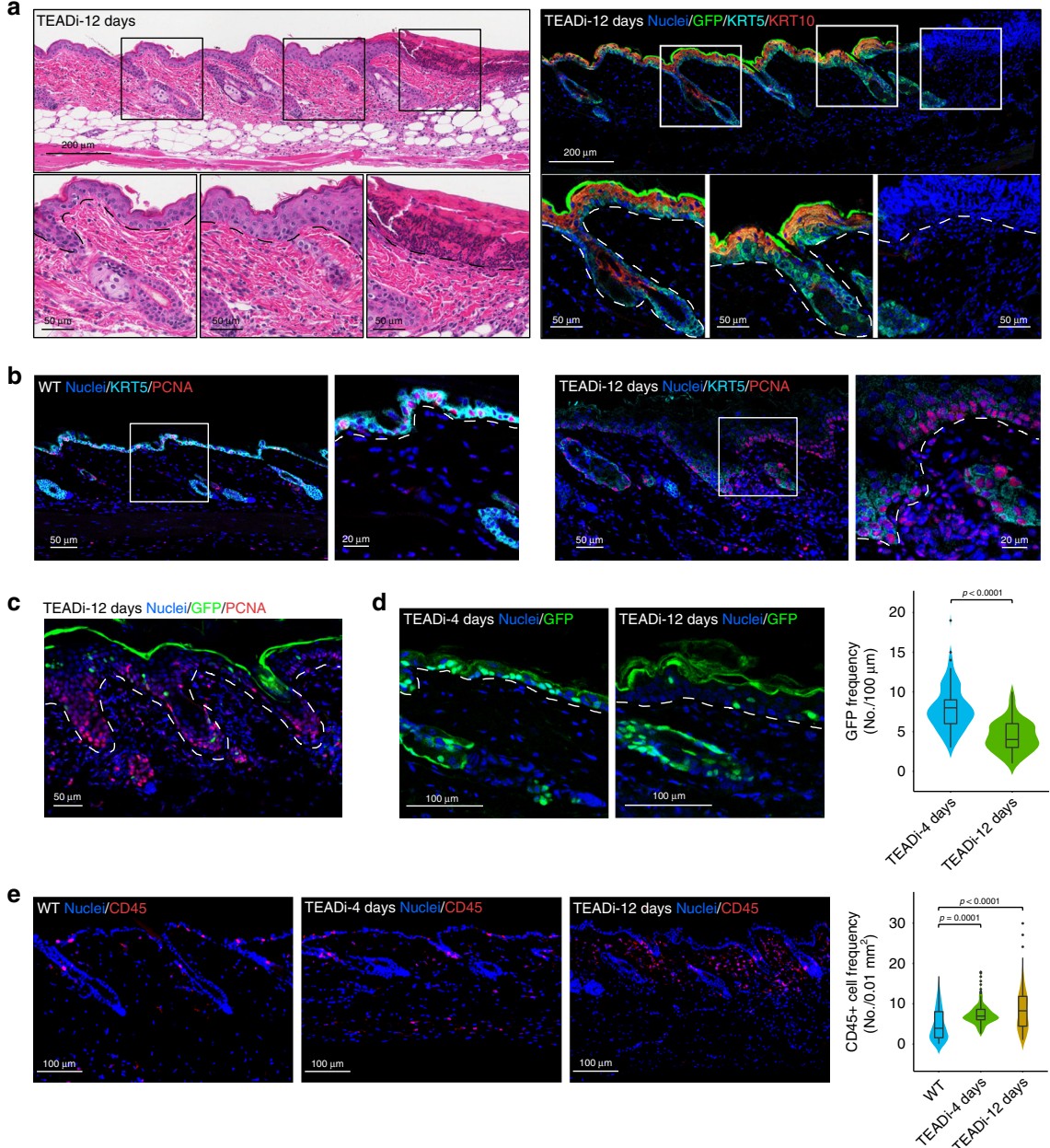

**Fig. 6 TEADi expression in the mouse skin induces wounding and inflammation. a** Histology and IF pictures of the skin of a TEADi-expressing mouse showing a wound area and the surrounding epithelium. **b** Representative IF pictures of the skin of WT or TEADi mice induced for 12 days to show expression of the proliferation marker PCNA and the basal marker KRT5. **c** Representative IF picture of the skin of TEADi mice induced for 12 days to show expression of the proliferation marker PCNA and GFP. **d** Representative IF pictures and quantification of the skin of TEADi mice induced for 4 and 12 days to show expression of GFP. **e** Representative IF pictures and quantification of immune cell infiltration labeled by CD45 staining in the skin of TEADi mice. Inserts show magnification of the highlighted area. Location of the basal membrane is indicated with a white dotted line. In **d** 4d $n = 69$ and 12d $n = 61$; (**e**) WT $n = 107$, 4d $n = 106$ and 12d $n = 114$ fields from 3 different mice in each condition. Violin plots shows density of data and box plot shows median, 25 and 75% quartile, 95% confidence interval and dots indicating potential outliers; in **d** two-tailed unpaired $t$ test and in **e** two-way ANOVA with Dunnett's multiple comparison test. Source data are provided as a Source Data file.

from Silvio Gutkind and has been described previously[10]. For KLF4 overexpression pMSCV-Flag-hKLF4 was used, a gift from Juan Belmonte (Addgene plasmid #20074).

**Cell culture, transfections, and adenoviral transductions**. All cells were cultured at 37 °C in the presence of 5% $CO_2$. HEK293 cells were obtained from AddexBio and Lenti-X™ 293T cells from Takara Bio and cultured in DMEM (Sigma-Aldrich Inc) containing 10% fetal bovine serum (FBS) (Sigma-Aldrich Inc) and antibiotic/antimycotic solution (Sigma-Aldrich Inc). N/TERT-2G keratinocyte cell line[24,25] was provided by Ellen H. van den Bogaard (Radboud University Medical Center, Nijmegen, The Netherlands) and James Rheinwald (Brigham and Women's Hospital, Boston, MA, USA), and cultured in EPILIFE medium (Life Technologies) with

Human Keratinocyte Growth Supplement (HKGS, Life Technologies). HEK293 cells and Lenti-X™ 293T were obtained directly from the described company and not further authenticated. N/TERT-2G cells were authenticated by STR profiling with the following results: TH01 8, 9.3; D5S818 11, 12; D13S317 8, 12; D7S820 9, 11; D16S539 10, 11; CSF1PO 10, 12; vWA 16, 18; TPOX 8, 11; Amelogenin X, Y. For siRNA experiments, cells were transfected with the corresponding siRNAs one day after plating and were treated/harvested 48 h after transfection. siRNAs were siGENOME SMARTpool from Dharmacon/Horizon, siE2F1 (M-003259-01-0005), siKLF4 (M-005089-03-0010), siYAP1 (M-012200-00-0005), siTAZ (M-016083-00-0005) and non-targeting control siRNA (D-001206-13). siRNA was transfected at a concentration of 8 pmol cm$^{-2}$ using Lipofectamine RNAiMAX (Invitrogen) according to the manufacturer's instructions. For adenoviral transduction, cells were

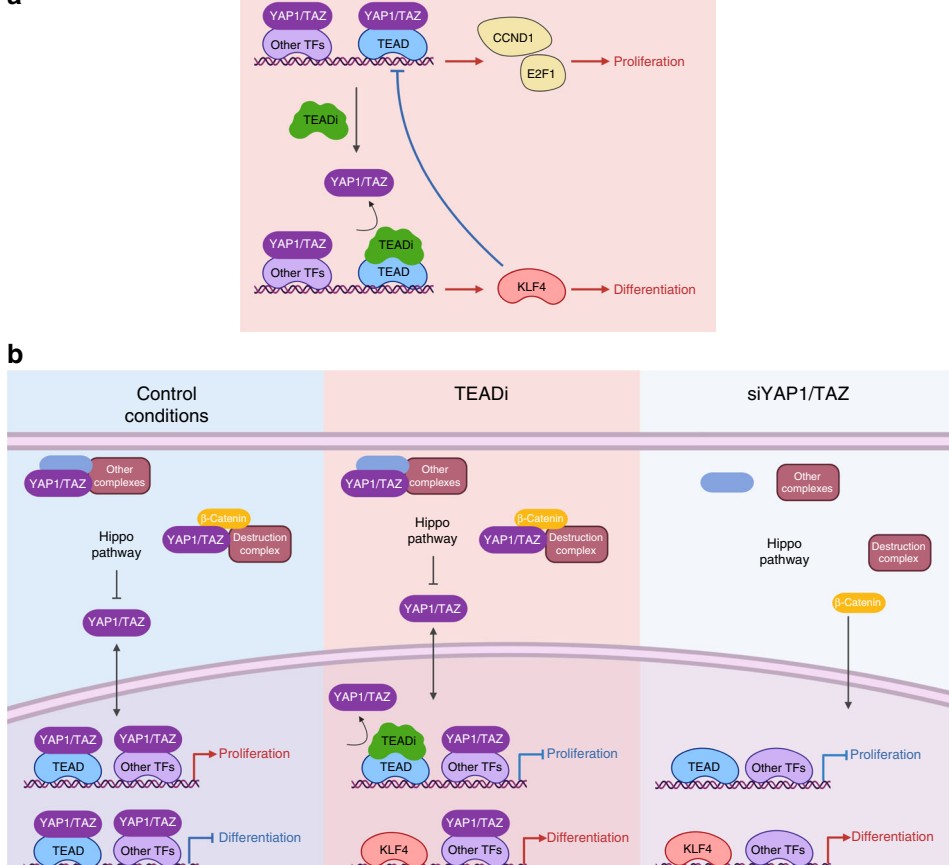

**Fig. 7 TEAD regulation of keratinocyte proliferation and differentiation. a** Simplified model summarizing the results from our study. Blockage of TEAD activation by TEADi triggers differentiation and reduced proliferation. In turn, activation of KLF4 can also lead to inhibition of YAP1/TAZ-TEAD transcription. **b** Comparison between the model of action of TEADi vs siYAP1/TAZ which might explain the lack of complete overlap in gene networks under both conditions. In cells expressing TEADi, YAP1/TAZ can still directly or indirectly regulate other factors that regulate transcription (TFs) and maintain cytoplasmic complexes, conditions that are not reproduced under YAP1/TAZ knockdown. Created with BioRender.

incubated with a multiplicity of infection (MOI) of 25 with adeno-GFP (control) or adeno-TEADi for the indicated times. TEADi adenoviruses were produced, purified and titered by Vector Biolabs in an adenoviral-Type 5 (dE1/E3) backbone with a CMV promoter, for GFP control Ad-CMV-GFP (Vector Biolabs, catalog no. 1060) was used. Lentiviruses and retroviruses were produced in Lenti-X 293T cells with the transfer and packaging plasmids, using TransIT-293 Transfection Reagent (Mirus) according to the manufacturer's instructions. Supernatant was collected 48 h after transfection and concentrated using Lenti-X Concentrator (Clontech, catalog no. 631232). Viruses resuspended in PBS were used to infect N/TERT-2G kerati-nocyte cell line overnight. Lenti-TRE-TEADi were produced by infecting N/TERT-2G keratinocytes or control GFP and selecting cells with G418 (SIGMA) for two weeks. Following selection, $1 \times 10^5$ cells were plated in cell culture inserts for organotypic culture as previously described[25] with minor modifications. Briefly, $1 \times 10^5$ cells were plated in 24 well plate cell culture inserts (Corning; catalog no. CLE3470-48EA) in EpiLife media supplemented with 50 μg/ml L-ascorbic acid (Sigma; catalog no. A4403) and 10 ng/ml keratinocyte growth factor (SIGMA; catalog no. K1757) for 48 h. Then, inserts were exposed to the air and media outside the insert was replaced with the supplemented media containing 1.5 mM $CaCl_2$ and 1 mg/ml of doxycycline to induce GFP and TEADi expression. The media was changed every 2 days and inserts were harvested after 11 days, fixed in Z-Fix, embedded in paraffin, and prepared for histological analysis. For proliferation analysis, cells were plated in 96 well plates, transduced with indicated constructs and proliferation was measured at indicated times using CellTiter-Glo Luminescent Cell Viability Assay (Promega) according to the manufacturer's instructions. To assess colony-forming efficiency[44], equal number of keratinocytes from corresponding mice were plated in triplicate in six-well plates and grown for 10–14 days. Kerati-nocytes from wild type (WT) or TEADi adult mice were isolated from back skin. Hair in the skin was clipped and skin was washed with 10% iodine twice and sterile PBS twice, and digested in trypsin 0.25% (GIBCO, #15050057) overnight at 4 °C. Then, the epidermis was scraped off the dermis and minced with scissors. Minced epidermis was filtered through a 100 μm cell strainer (Corning, #352360) and cells were centrifuged at $400 \times g$ for 8 min, resuspended in keratinocyte media described above and plated in collagen coated plates.

**Gene expression analysis and quantitative PCR**. Keratinocyte RNA was isolated and processed using RNeasy Plus Mini Kit (Qiagen) according to the manu-facturer's instruction. Cells were lysed using the Precellys lysing kit (Bertin Instruments). mRNA integrity was measured with Agilent TapeStation system and samples with RIN < 8 were not further processed. mRNA expression profiling was performed in the CCR-Sequencing Facility at the NIH. Reads of the samples were trimmed for adapters and low-quality bases using Trimmomatic software before alignment with the reference genome Human—hg19 and the annotated transcripts using STAR. Gene counts were filtered by genes with ≥8 reads and normalized to TMM (Trimmed Mean of M values) and TPM (Transcripts Per Million) using Partek Flow software, version 7.0 2018 (Partek Inc). TMM normalized counts were used for differential analysis using PARTEK Flow GSA algorithm (Partek Inc). Unless indicated otherwise, Gene Ontology (GO) terms were obtained with ToppGene[45] using genes with $q < 0.05$ and fold change ≥1.5 for upregulated genes and ≤−1.5 for downregulated genes. Canonical pathways and upstream regulators analysis were generated with Ingenuity Pathway Analysis (IPA, Ingenuity Systems, www.ingenuity.com) by using genes with $q < 0.05$ and $|FC| \geq 1.5$. Analysis of over-represented conserved transcription factor binding sites was performed with oPOSSUM[46] using upregulated ($q < 0.05$, FC ≥ 1.5) and downregulated ($q < 0.05$, FC ≤ −1.5) genes by TEADi expression.

One microgram of cDNA was used as template for quantitative polymerase chain reaction with reverse transcription (qRT–PCR) analysis using SensiFAST cDNA Synthesis Kit (Bioline) and SensiFAST SYBR Hi-ROX Kit (Bioline). Samples were analyzed using a 7900HT Fast Real-Time PCR System. Oligonucleotides used for amplification were (Gene, Forward sequence 5′→3′, Reverse sequence 5′→3′): *RPLP0*, 5′-TGTCTGCTCCCACAATGAAAC-3′, 5′-TCGTCTTTAAACCCTG CGTG-3′; *KRT5*, 5′- ATCTCTGAGATGAACCGGATGATC-3′, 5′-CAGATTG

GCGCACTGTTTCTT-3′; *KRT1*, 5′-CTGAGCTGAATCGTGTGATCC-3′, 5′-GCT TGTTCTTGGCATCCTTG-3′; *TGM3*, 5′-GGAAGGACTCTGCCACAATGTC-3′, 5′-TGTCTGACTTCAGGTACTTCTCATACTG-3′; *CTGF*, GCTCGGTATGTCTT CATGCTG, GAAGCTGACCTGGAAGAGAAC; *CYR61*, TGGAGTTATATTCAC AGGGTCTG, GCAGCTCAACGAGGACTG.

**Immunoblot analysis.** For Western blot[44,47] cells were lysed by sonication at 4 °C in lysis buffer (50 mM Tris-HCl, 150 mM NaCl, 1 mM EDTA, 1% Nonidet P-40, 0.5% sodium deoxycholate, 0.1% SDS) supplemented with complete protease inhibitor cocktail (Roche, #6538304001) and phosphatase inhibitors (PhosSTOP, Sigma-Aldrich, #4906837001). Equal amounts of total cell lysate proteins were subjected to SDS-polyacrylamide gel electrophoresis and transferred to PVDF membranes. Antibodies used were: anti-GAPDH (Cell Signaling; clone no. 14C10; catalogue no. 2118; 1:2000), anti GFP (Cell Signaling; clone no. D5.1; catalogue no. 2956; 1:2000), anti-HA tag antibody (Cell Signaling; clone no. C29F4; catalogue no. 3724; 1:1000), anti-FLAG tag antibody (Cell Signaling; clone no. 9A3; catalogue no. 8146; 1:1000), Myc tag antibody (Cell Signaling; clone no. 71D10; catalogue no. 2278; 1:1000), YAP1 (Cell Signaling; clone no. D8H1X; catalogue no. 14074; 1:1000), TAZ (Cell Signaling; clone no. V386; catalogue no. 4883; 1:1000), LATS1 (Cell Signaling; clone no. C66B5; catalogue no. 3477; 1:1000), Pan-TEAD (Cell Signaling; clone no. D3F7L; catalogue no. 13295; 1:1000), p63 (Cell Signaling; clone no. D9L7L; catalogue no. 39692; 1:1000), KRT5 (BioLegend; catalog no. 905901; 1:1000), KRT10 (BioLegend; catalogue no. 905401; 1:1000), Involucrin (SantaCruz; catalogue no. sc-21748; 1:500), E2F1 (Cell Signaling; catalogue no. 3742; 1:1000), Cyclin D1 (Cell Signaling; clone no. 92G2; catalogue no. 2978; 1:1000), Cyclin E1 (Cell Signaling; clone no. D7T3U; catalogue no. 20808; 1:1000), p53 (DAKO; clone no. DO-7; catalogue no. GA616; 1:1000), p16 (SantaCruz; catalogue no. sc-56330; 1:200), KLF4 (Cell Signaling; clone no. D1F2; catalogue no. 12173; 1:1000), CYR61 (Cell Signaling; clone no. D4H5D; catalogue no. 14479; 1:1000), GAL4 (SantaCruz; catalogue no. SC510; 1:1000), Total β-Catenin (Cell Signaling; clone no. D10A8; catalogue no. 8480; 1:1000), Active β-Catenin (Cell Signaling; clone no. D2U8Y; catalogue no. 19807; 1:1000), Cleaved-Caspase3 (Cell Signaling; clone no. 5A1E; catalogue no. 9664; 1:1000), Cleaved-PARP (Cell Signaling; clone no. D64E10; catalogue no. 5625; 1:1000). Secondary HRP-conjugated antibodies used were: Pierce peroxidase goat antimouse IGG (H + L) (ThermoFisher, catalogue no. 31432; 1:4000), Pierce peroxidase goat antirabbit IGG (H + L) (ThermoFisher, catalogue no. 31462; 1:4000), Anti-rat IgG (Cell Signaling; catalogue no. 7077; 1:4000), Mouse Anti-rabbit IgG Conformation Specific (Cell Signaling, clone no. L27A9; catalogue no. 5127; 1:4000) and Rabbit Anti-Mouse IgG Light Chain Specific (Cell Signaling, clone no. D3V2A; catalogue no. 58802; 1:4000). Secondary antibody was incubated at RT for 1 h. Bands were detected using a ChemiDoc Imaging System (Bio-Rad) with Clarity Western ECL Blotting Substrates (Bio-Rad) according to the manufacturer's instructions. Blot images were processed and quantified using ImageLab software v5.2.1 (Bio-Rad). Uncropped scans are available in the Source Data file.

**Luciferase assays and immunoprecipitation.** Luciferase assays were performed in HEK293 cells. To measure TEAD activity, cells in 12 or 24 well plates were co-transfected overnight with 8xTEAD-Luc (0.25 μg cm$^{-2}$) plus the indicated DNA constructs: GFP (0.4 μg cm$^{-2}$), TEADi (0.4 μg cm$^{-2}$), Mock Control (0.2 μg cm$^{-2}$), YAP1 (0.2 μg cm$^{-2}$), TAZ (0.2 μg cm$^{-2}$). Next day cells were serum starved overnight and then luciferase activity was measured using a Dual-Glo Luciferase Assay Kit (Promega) and a Microtiter plate luminometer (SpectrMax iD3, Molecular Devices LLC). To measure GAL4-KLF4 activity, cells were co-transfected with GAL4-KLF4 constructs (0.3 μg cm$^{-2}$), UASLuc (0.2 μg cm$^{-2}$) plus the indicated DNA constructs at 0.3 μg cm$^{-2}$ and processed as described above. Luciferase normalization was performed in every case by co-transfecting a Renilla Luciferase Vector (0.025 μg cm$^{-2}$) (Promega). For immunoprecipitation (IP), the following antibodies were used: anti-HA tag antibody (Covance; clone no. 16B12; catalogue no. MMS-101R; 1:100), anti-FLAG tag antibody (Biolegend; clone no. L5; catalogue no. 637304; 1:100), YAP1 (Cell Signaling; clone no. D8H1X; catalogue no. 14074; 1:100), Pan-TEAD (Cell Signaling; clone no. D3F7L; catalogue no. 13295; 1:100), GAL4 (SantaCruz; catalogue no. SC510; 1:100), KLF4 (Cell Signaling; clone no. D1F2; catalogue no. 12173; 1:100). Proteins were extracted in IP lysis buffer (50 mM Tris-Cl pH 7.5, 150 mM NaCl, 10% glycerol, 2 mM MgCl$_2$, and 1% NP40) supplemented with a complete protease inhibitor cocktail (Roche, #6538304001). Protein extracts were subjected to centrifugation at 14,000 rpm for 10 min and then supernatant protein was quantified. Equal amounts of protein were incubated with primary antibodies at 4 °C overnight with Dynabeads Protein G (Invitrogen, #10004D). Then, beads were washed 4 times with IP lysis buffer and proteins were eluted with denaturing loading buffer for western blot analysis.

**Mice.** All animal studies were carried out according to approved protocols from the NIH-Intramural Animal Care and Use Committee (ACUC) of the National Cancer Institute, in compliance with the Guide for the Care and Use of Laboratory Animals. The generation of TRE-TEADi transgenic mice was performed with the assistance from the CCR Transgenics Facility. TEADi coding sequence was cloned downstream of the seven tet-responsive element (TRE) in a modified inducible vector containing a pTIGHT inducible promoter, a woodchuck hepatitis virus posttranscriptional regulatory element (WPRE) and a poly(A) sequence, all flanked by the chicken h-globin gene insulator (HS4) sequence to avoid positional effects and transgene silencing[48]. The fragment containing the expression cassette was isolated by PmeI digestion from vector DNA and purified for microinjection into C57 mice fertilized oocytes. Founders were identified for the presence of the transgene by screening genomic DNA from tail biopsies using a PCR reaction with the following primers: forward sequence 5′ CGCGTTAAGTGCAACACGAT 3′, reverse sequence 5′ GAGAAACACTGGACGCCGTA 3′, band approximately 250 bp. PCR reactions were performed as follow: 95 °C for 5 min, followed by 30 cycles of 95 °C for 30 s, 60 °C for 30 s, and 72 °C for 30 s, and a final cycle with 5 min of extension at 72 °C. FVB/N mice carrying the cytokeratin 5 promoter in the reverse tetracycline trans-activator (rtTA) (KRT5-rtTA) were provided by Silvio Gutkind[36]. Both male and female mice were used in the studies and all experiments were conducted using littermate controls. Housing conditions were as follow: temperature set point is 72 ± 4 °F (22.2 ± 2.2 °C), light cycle of 12 h on 6 am to 6 pm and 12 h off, NIH-03I rodent diet. Doxycycline was administered in the food grain-based pellets (Bio-Serv) at 6 g kg$^{-1}$. Doxycycline treatment was started between weeks 6 and 10 after birth.

**Immunofluorescence.** For IF in cells, cells were seeded in coverslips or 24 well plates and treated as described. Then cells were washed with ice-cold PBS and fixed with 3.2% paraformaldehyde in PBS for 10 min at room temperature. After washing three times with PBS, cells were permeabilized with TritonX100 0.1% in glycine 200 mM in PBS and nonspecific binding was blocked with 3% of bovine serum albumin (BSA) in PBS for 1 h. Fixed cells were incubated with the primary antibody overnight at 4 °C, washed three times with PBS and then incubation with the secondary antibody for 1.5 h at room temperature. Immunofluorescence analysis of mouse skin was performed on tissue sections embedded in paraffin. Sections were prepared for staining by antigen retrieval in 10 mM Sodium Citrate buffer pH = 6, washed and blocked with 3% BSA for 1 h at room temperature. Slides were then incubated with the primary antibody overnight at 4 °C, washed three times with PBS and then incubation with the secondary antibody for 1.5 h at room temperature. Sections were mounted in FluorSave Reagent (Millipore, #345789) with #1.5 coverslips for imaging. Nuclei were stained with Hoechst 33342 (1:2000, Invitrogen, #H3570). The following antibodies were used: GFP (Cell Signaling; clone no. D5.1; catalogue no. 2956; 1:500), GFP Polyclonal Antibody (ThermoFisher; catalogue no. A-6455; 1:500), GFP Polyclonal Antibody Alexa Fluor 488 (ThermoFisher; catalogue no. A-21311;1:500), Pan-TEAD (Cell Signaling; clone no. D3F7L; catalogue no. 13295; 1:200), KRT10 (BioLegend; catalogue no. 905401; 1:400), PCNA (Cell Signaling; catalogue no. 13110S; ice-cold methanol fixation; 1:400), KRT5 (BioLegend; catalog no. 905901 and 905501; 1:200), p63 (Cell Signaling; catalog no. 39692S; 1:400), CYR61 (Cell Signaling; clone no. D4H5D; catalogue no. 14479; 1:100), YAP1 (Cell Signaling; clone no. D8H1X; catalogue no. 14074; 1:200), Phospho-Rb (Ser807/811) (Cell Signaling; clone no. D20B12; catalogue no. 8516; 1:1000), human KLF4 (Cell Signaling; clone no. D1F2; catalogue no. 12173; 1:100), mouse KLF4 (R&D; catalogue no. AF3158; 1:200), Loricrin (BioLegend; catalogue no. 905101; 1:100), CD45 (Cell Signaling; clone no. D3F8Q; catalogue no. 70257; 1:100). The secondary antibodies were incubated at room temperature for 2 h Donkey anti-Rabbit IgG Alexa Fluor 555 (Invitrogen; catalog no. A-31572; 1:1000), Goat anti-Chicken IgY Alexa Fluor 633 (Invitrogen; catalog no. A-21103; 1:1000), Donkey anti-Goat IgG Alexa Fluor 546 (Invitrogen; catalog no. A-11056; 1:1000). EdU staining was processed following the manufacturer's instructions of Click-IT EdU Imaging Kit (Invitrogen) with 4 h EdU incubation. Images were obtained using a Leica SP8 confocal microscope with LASX software (equipped with either a high-contrast Plan-Apochromat 20x oil CS2 objective at 1.30 NA; Leica Microsystems) or with a Keyence BZ-X700 with automatic stage and focus with BZX software (Equipped with a CFI Plan Apo λ20x NA 0.75, Nikon). Quantification of the expression was done in the BZX analysis software (Keyence) equipped with hybrid cell count and macro cell count. Different fields per sample were analyzed at the same counting conditions by the software automatically. For Fig. 5c–f and Supplementary Fig. 1h, i, quantification was performed in stitched fields by selecting the epithelia region on the basis of its histological appearance. In Fig. 6d, quantification was performed in individual fields by selecting the epithelia region on the basis of its histological appearance. In Figs. 3b and 6e, Supplementary Fig. 1a–f and 1d, quantification was performed in individual fields. The expression level was calculated based on the area and intensity of each positive signal. Final images were bright contrast adjusted with LASX software (Leica), BZX analysis software (Keyence) or PowerPoint. For histological analysis, tissues were embedded in paraffin and 3-μm sections were stained with H&E. Stained H&E slides were scanned at 40x using an Aperio CS Scanscope (Aperio).

**Statistics and reproducibility.** All analyses were performed in triplicate or greater and the means obtained were used for ANOVA or independent t-tests. Statistical analyses, variation estimation and validation of test assumptions were carried out using the Prism 7 statistical analysis program (GraphPad). Statistical analysis of intersections in Venn diagrams was performed by hypergeometric test (one tailed Fisher's exact test). Asterisks denote statistical significance. For the following figures, the picture/blot shown is representative from this number of independent experiments: Fig. 1a n = 2; 1c *n* = 3; 1f *n* = 2; 1g *n* = 3; 1h *n* = 3; 2a *n* = 2; 2g *n* =

5; 2j $n = 3$; 2l $n = 2$; 3c $n = 3$; 3d $n = 2$; 3e $n = 2$; 3k $n = 5$; 4a $n = 3$; 4b $n = 2$; 4d $n = 2$; 4f $n = 3$; 5b $n =$ at least 7 mice for each condition; 6a $n = 3$ mice; 6b $n = 3$ mice for each condition; 6c $n = 3$ mice; S1g $n = 2$; S2a $n = 3$; S2f $n = 2$; S4a $n = 3$ mice for each condition; S4b $n = 2$ mice with 3 technical replicates each.

**Reporting summary**. Further information on research design is available in the Nature Research Reporting Summary linked to this article.

## Data availability

The authors declare that all data supporting the findings of this study are available within the article and its Supplementary Information files or from the corresponding author upon reasonable request. RNAseq primary and processed data generated in this manuscript have been deposited in the GEO database under accession codes: GSE136876, GSE137410, and GSE137531. Processed RNAseq data is provided in Supplementary Data 1–3. The source data underlying Figs. 1b–e, f, 2a, g, j–l, 3b–f, 3k, 4a–f, 5c–f, 6d, e, and Supplementary Figs. 1a–i, 2a–c, 2f, g, 2i, j, 3a–d, are provided as a Source Data file.

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

## Acknowledgements

This research was supported by the Intramural Research Program of the National Institutes of Health, National Cancer Institute, Center for Cancer Research (ZIA BC 011764 and ZIA BC 011763). Yao Yuan and Qianming Chen were supported by grants from the National Natural Science Foundation of China (81520108009, 81621062), and 111 Project of MOE (B14038), China. This work used the computational resources of the NIH High-Performance Computing Biowulf Cluster. We thank the members of the CCR

Sequencing Facility at the Frederick National Laboratory for Cancer Research for their help during sample preparation, sequencing and data processing.

## Author contributions

R.I.B. initiated the study; Y.Y. and R.I.B. designed the study and experiments; Y.Y., J.P., A.F., and R.I.B. performed experiments; P.A. generated TRE-TEADi transgenic mice; Y.Y., J.P., A.F., and R.I.B. analyzed and interpreted data; Z.W., Q.C., and R.I.B. provided administrative, technical, and material support; Y.Y. and R.I.B. prepared the paper.

## Competing interests

The authors declare no competing interests.
