## [Peer Review File · Nature Communications]

Reviewers' Comments:

Reviewer #1:

Remarks to the Author:

The authors constructed a genetically-encoded inhibitor of YAP/TAZ-TEAD interaction (TEADi) and showed it localizes to the nucleus and can repress YAP and TAZ-mediated transcriptional activity. TEADi binds YAP and TAZ and prevents TEAD interaction. They next expressed TEADi in immortalized keratinocytes and showed both in 2D culture and 3D assays that it caused a strong reduction in basal/progenitor marker expression, and increase in differentiation markers, reduced CTGF and CY61 expression and decreased proliferation. Did RNAseq on N/TERT2G keratinocytes expressing TEADi and identified several genes regulated by YAP/TAZ-TEAD. Among those was E2F1 and they confirmed their RNA seq data by showing that E2F mRNA and protein levels. They found TEAD1 reduced the activity of a reporter construct driven by part of the E2F promoter. RNAi against E2F reduced keratinocyte proliferation. Analysis of the potential Transcription factor binding sites in upregulated genes showed a clear enrichment for KLF4. They next found that TEADi increased KLF4 nuclear expression. Klf4 was necessary for TEADi-induced differentiation and loss of KLF4 increased the expression of CYR61 and the interaction of YAP/TEAD (but not TAZ/TEAD) whereas KLF4 expression reduced it. They next explored the effects of TEADi in vivo using a mouse model in which TEADi expression was targeted to the epidermis using a tetracycline inducible transactivator driven by the K5-promoter. TEADi expression caused disrupted skin homeostasis marked by increased differentiation markers and decreased proliferation. The epidermis also showed increased KLF4 expression. Lastly they expressed TEADi in SCC cells and saw a decrease in proliferation, but no induction of differentiation. Analysis of gene expression changes showed that there was partial overlap with the changes seen in primary keratinocytes, including KLF4 and several genes with KLF4 binding sites in their promoter.

This is a well-written and very nice paper that will certainly be of interest to the YAP/TAZ-Hippo field and likely a broader audience interested in epidermal biology and disease. The findings are novel and the impactful. TEADi is seemingly a very powerful reagent that is long overdue and will be extremely useful to researchers in the field. The description of the YAP/TAZ-TEAD-mediated regulation of KLF4 and its role in skin homeostasis are also very interesting. The data are of high quality and individual experiments appear to include the appropriate controls. However, there are some concerns that need to be addressed.

1. The TEADi reagent contributes significantly to the impact and novelty of this work. The authors clearly demonstrate that TEADi disrupts the interaction between TEADs and YAP or TAZ, and that it impairs YAP/TAZ function and the expression of YAP/TAZ target genes. This suggests that TEADi is a potent YAP/TAZ-TEAD inhibitor. However, an important remaining question is does TEADi have any YAP/TAZ-independent effects in cells? The authors provide no data testing this possibility and seem to conclude that all observed effects of TEADi are due to loss of YAP/TAZ-TEAD-function. Does TEADi influence gene expression in YAP/TAZ knockdown cells? RNAseq on cells with Yap/TAZ KD + TEADi seem the best way to conclusively test this, but minimally they should confirm that TEADi does not alter the expression of key target genes in cells that lack YAP and TAZ (see #2 below).

2. A related concern is that the authors conclude that the observed effects of TEADi on keratinocyte differentiation, proliferation, and gene expression are due to loss of YAP/TAZ-TEAD function. Though a logical and likely conclusion, they do not rule out the possibility that TEADi promotes differentiation or KLF4 expression through a YAP/TAZ-independent mechanism. Some in vitro experiments that demonstrate that TEADi does not promote the same changes in differentiation markers, proliferation, KLF4 and some key KLF4 target genes in YAP/TAZ KD cells should address this concern. One would also expect YAP/TAZ KD to at least partially phenocopy TEADi expression. These experiments could also help address concern #1.

3. The schematics in Figure 3H and Figure 5E and their conclusions appear to suggest that

disruption of YAP/TAZ-TEAD interaction promotes YAP (TAZ?) interaction with KLF4 and the recruitment of YAP or TAZ to the promoters of KLF4-regulated genes. The authors show that KLF4 interacts with YAP, TAZ, and TEAD, and that TEADi increases this interaction (though not the TAZ-KLF4/interaction). Though consistent with their model, these data alone are not sufficient to conclude this. It remains possible that the reduced YAP/TAZ-TEAD interaction alters KLF4-mediated transcription through an indirect mechanism or through altered expression of some other co-factor. If the authors want to conclude that a YAP/KLF4 complex is driving transcription and that this is enhanced by loss of YAP-TEAD interaction, more data is necessary. For example, is YAP recruited to the promoters of KLF4 target genes and does TEADi enhance this? Does disruption of YAP/KLF4 interaction impair the expression of key KLF4 regulated genes and keratinocyte differentiation? Alternatively, the authors could discuss this as one potential model that is supported by their data, but that requires additional investigation.

4. Though the data are convincing, it is not clear the appropriate statistical tests were used. In several figures the legend suggests that a T-test was used to establish significance, yet many panels include multiple comparisons and neither the legend nor the methods indicate whether a correction for multiple comparisons was done (for example T-test with Bonferroni correction).

Reviewer #2:

Remarks to the Author:

The authors have engineered a chimeric protein, TEADi, that is able to compete with and thus inhibit TEAD function in cells, thus selectively inhibiting the fraction of YAP/TAZ activity that is TEAD-dependent. They then combined cell-based and tissue-based assays, the latter including sophisticated transgenic mouse models, to show that TEAD inhibition profoundly disrupts cell cycle regulation as well as terminal differentiation. The latter aspect is significantly related to the misregulation of KLF4, a known transcriptional effector of terminal differentiation in the epidermis *in vivo*. The authors provide a novel take on the interplay between effectors of Hippo signaling and KLF4, a known driver of terminal differentiation in the epidermis. Finally, they provide evidence that at least part of the TEAD-dependent molecular network is preserved in human tumor epithelial cell lines but that the associated cellular physiological consequences are very different. The tumor biology-related aspect of the manuscript is comparatively weak and could well be preserved for a future contribution.

This is a substantive, original, important and beautiful study that should be published. I do have comments and suggestion which I believe to be reasonable, and I am confident that the authors can and will consider and address them.

Fig 1a / How big is TEADi? This should be related in the main text. An effective strategy would be to provide amino acid landmarks for the various domains making up TEADi, as shown in Fig 1a. Also, perhaps the authors should enhance their molecular account of how they think TEADi functions.

Fig 1h / The authors should quantitate the frequency of K10 positive cells in the control and TEADi-expressing epidermal equivalents.

Fig 2 / The authors should make available their entire RNAseq data set.

Fig 2c / The overlap in TEAD-sensitive genes (n=4) with other/previous studies is extremely limited (4 out of a potential of >1500 ORFs – and no statistical test being reported, unlike other instances of similar comparisons elsewhere in the ms.). Why is it so limited, and is the main conclusion actually supported by the data?

p 6 / The authors state: "revealing that cell cycle exit and activation of differentiation programs in

keratinocytes are not concomitantly activated". They should explore the literature for precedents, and report on them.

p 6 / The authors conclude: "...and that cell cycle exit is not responsible for the activation of differentiation programs." It appears that they don't have enough evidence to make the latter aspect of the conclusion "conclusive".

Fig 3c / The authors limit their WB analysis for differentiation product to K10 – it would be very desirable to test for additional markers if they wish to maintain a claim about a global effect on keratinocyte differentiation as opposed to a specific effect on K10 expression. Also, correct the CRY61 erroneous label in this figure.

p 8 / (minor) I would refrain from using the term "hyperplasia" to describe what's happening in these compensated epidermal areas – I would simply use the term "thickened epidermis". Hyperplasia is usually used to convey enhanced proliferation.

p 8 (major) I think the authors really need to provide additional evidence for their account of the cellular mechanism(s) that account for the thickened epidermis in these compensated regions. I am not aware of any precedent for this. This is a very weak spot in the manuscript.

p 9 Fig 6 / Again here the authors should show make their entire RNA seq data set available.

p 9 Fig 6 / The authors should provide substantive evidence that SCC15 and SCC13 cells are actually capable of a significant degree of differentiation before concluding about the uncoupling of the impact of TEADi on cell cycle vs. terminal differentiation. What are their positive controls for substantive differentiation of these cell lines? This concern is further fueled by the last paragraph of the results section. I think the data supporting the maintenance of a link between KLF4 function and TEADi is likely correct – but I would recommend a more prudent interpretation of the implications for the regulation of terminal differentiation (or lack thereof) in aberrant models such as SCC13 and SCC15...

Fig 6 / The statistical p value for overlap ($p < 0.05$) is borderline.

A prediction from the data herein reported is that KLF4 regulation and function may well be mechanosensitive in the epidermis. The authors should comment on this.

Reviewer #3:

Remarks to the Author:

In this manuscript, Yuan et al. use a genetically-encoded inhibitor of the interaction between YAP1/TAZ and TEAD (TEADi) to dissect the specific regulatory gene expression networks in keratinocytes and mouse skin. The authors confirmed previous observations made by others that show that YAP1/TAZ-TEAD regulate epithelial stem cell homeostasis and found a regulatory loop with KLF4 important for cell differentiation. While the manuscript has some interesting observations regarding YAP-TEAD-KLF4 interaction and has developed a genetic tool for in vivo and in vitro studies, most findings are not entirely novel and the development of TEADi is not well justified nor tested in a way that would be required to prove its advantage over knock out studies. Thus, apart from other relatively minor problems, I have three major concerns:

1. Lack of significant novelty: The observation that a TEAD dependent genetic program is required for skin homeostasis has been made before in a number of papers that have gone deeper than the current manuscript.

2. The necessity and usefulness of the TEADi is poorly justified. Many studies have successfully

used YAP;TAZ double knock out strategies, so this cannot be it. Also, the authors argue that TEADi would circumvent the problem of disrupting potential YAP/TAZ function in the cytoplasm and interactions with other transcription factors. However, fail to make a convincing case for their argument. They do not show whether TEADi and YAP;TAZ double knockout indeed give different results. Without such comparison their argument that TEADi provides an advantage over YAP;TAZ knockout is fantasy.

3. While this study is nicely illustrated, it is shallow and in places naïve and intellectually ignorant. For example, TEADs are known to interact with VGLL4, which counteracts Yap, Taz. However, there is no mention of this let alone a test whether this interaction is affected, nor an argument how this may make TEADi different from YAP;TAZ knockout. Given that the YAP/TAZ-TEAD interaction inhibitor also inhibits Vgl4-TEAD interaction, how can these phenotypes be entirely attributed to YAP/TAZ inhibition? How do these phenotypes compare to when VGLL4 is overexpressed and knocked out in basal cells? Also, the analysis of gene expression profiles too superficial to be at a level required for a publication with impact. For example, the authors compare their profile with that of two other studies (Zhao and Zanconato). They simply say there is significant overlap. However, I find the overlap between these three studies surprisingly minimal. This is especially true between the other two published studies, where only a fraction of enriched genes overlap. I am thus surprised how the authors interpret these data, it seems to me that wishful thinking and biased expectations were dictating the argument rather than a careful analysis of the results. I thus think that the authors really missed a chance here and in other parts of their study to truly figure out something novel, rather than putting together an poorly supported argument that does not even provide any novelty.

Other:

How does KLF4 regulate the YAP-TEAD mediated CYR61 expression? Does KLF4 overexpression or KD affect YAP-TEAD binding? Co-IP of YAP-TEAD in presence/absence of KLF4 and its repressor/activator variants (1-566 and 159-329).

Does dox inducible TEADi reduced basal cell proliferation or caused their elimination? Are basal cells positive for TUNEL and cCasp3?

What happens with cell proliferation and cell death in the epidermis reconstruction assay stably expressing a tetracycline-inducible TEADi (Fig1G)?

TEADi induction caused epidermal hyperplasia around wounds (Fig. 4b and 4c). Are these hyperplastic areas expressing TEADi and thus negative for YAP activity? How could these YAP repressed cells proliferate, but basal cells show reduced proliferation? In particular, if the authors showed that TEADi leads to a downregulation of cell cycle processes in both cancer and normal cells, how could a tissue become hyperplastic?

NCOMMS-19-16447-T Response to Reviewers' comments:

Reviewers comments are in italic

Reviewer #1:

1. The TEADi reagent contributes significantly to the impact and novelty of this work. The authors clearly demonstrate that TEADi disrupts the interaction between TEADs and YAP or TAZ, and that it impairs YAP/TAZ function and the expression of YAP/TAZ target genes. This suggests that TEADi is a potent YAP/TAZ-TEAD inhibitor. However, an important remaining question is does TEADi have any YAP/TAZ-independent effects in cells? The authors provide no data testing this possibility and seem to conclude that all observed effects of TEADi are due to loss of YAP/TAZ-TEAD-function. Does TEADi influence gene expression in YAP/TAZ knockdown cells? RNAseq on cells with Yap/TAZ KD + TEADi seem the best way to conclusively test this, but minimally they should confirm that TEADi does not alter the expression of key target genes in cells that lack YAP and TAZ (see #2 below).

The peptides used to build TEADi were selected due to their proven specificity towards TEAD binding in previous reports^{1,2,3}, and we expanded the Results section of “Development of a genetically-encoded YAP1/TAZ-TEAD interaction inhibitor” to further explain this point.

Nevertheless, we understand the concern of the Reviewers that despite this reported specificity of TEAD binding domains (TBDs) towards YAP1/TAZ-TEAD inhibition, possible additional nuclear effects of the TBDs used cannot be completely disregarded. In order to further corroborate the specificity of the effects towards YAP1/TAZ-TEAD signaling, we performed the following experiments:

- To further compare the transcriptional consequences of TEAD blockage with those triggered by removal of YAP1 and TAZ, we performed RNA-seq in N/TERT2G keratinocytes with knockdown of YAP1/TAZ (siYAP1/TAZ, new Supplementary Table 2, new Supplementary Figure 2a). As expected, siYAP1/TAZ produced a larger amount of significant differentially-regulated genes compared with TEADi, probably due to additional transcriptional partners and disruption of cytoplasmic functions of these proteins (new Fig. 2d). However, significant overlap was observed between differentially regulated genes in both conditions (new Fig. 2d), confirming that common gene networks are present in TEADi and siYAP1/TAZ datasets. We believe that the lack of complete overlap of these datasets could be due to several factors that include: the mode of action and timing of a protein inhibitor (TEADi) versus a gradual decrease in protein expression (siRNA); the fact that siRNA does not completely deplete YAP1 and TAZ protein expression (new Supplementary Figure 2a); the fact that in cells expressing TEADi, YAP1 and TAZ could interact with other proteins and transcription factors, a condition that is not present in cells transduced with siRNA; and the transcriptional pathways affected by TEAD versus siYAP1/TAZ are not completely overlapping (see below example on Wnt signaling).
- Analysis of gene sets common to TEADi and siYAP1/TAZ, which would constitute the core of YAP1/TAZ-TEAD transcriptional activity, showed similar GO terms related to cell cycle and differentiation (new Supplementary Fig. 2b) to those observed in TEADi conditions.
- Side by side analysis of the expression of several markers in TEADi and siYAP1/TAZ conditions showed consistent global changes in gene regulation, particularly on key markers like *CTGF*, *CYR61*, *IVL*, *FOXN1*, *KRT1* and *KRT10* (new Fig. 3h).
- We performed the suggested experiments combining siYAP1/TAZ with TEADi. We found that siYAP1/TAZ leads to an increase in KLF4 mRNA and protein levels, activation of the differentiation marker KRT10, and downregulation of CYR61, which recapitulates the effect of TEADi but is not further increased by the expression of the TEAD inhibitor (new Fig. 3i).
- Another way to probe the specificity of TEADi is to test if this inhibitor can distinguish between YAP1/TAZ TEAD-dependent and independent events. Regulation of the activation of β -catenin has been shown to be independent of TEAD transcription factors and dependent on the direct interaction of YAP1

and TAZ with components of the Wnt signaling pathway^{4,5}, therefore we tested whether TEADi could be used to distinguish between YAP1/TAZ and TEAD specific events on Wnt signaling. Analysis of canonical pathways affected by siYAP1/TAZ and TEADi in Ingenuity Pathway Analysis (IPA) indicated that, although under both conditions Wnt/ β -catenin signaling gene networks are differentially regulated (new Fig. 2e), siYAP1/TAZ leads to a clear increase in the activity of the pathway when compared with TEADi (new Fig. 2f). Indeed, several downstream targets of Wnt signaling were activated by siYAP1/TAZ and not TEADi, including *AXIN2* and *CD44* (new Fig. 2e). By utilizing an antibody that recognizes non-phospho (active) β -Catenin (Ser45)⁶, we confirmed that siYAP1/TAZ results in an increase in the amount of active β -catenin in keratinocytes, while TEADi results in minimal alterations in the levels of active protein (new Fig. 2g). Our results indicate that YAP1/TAZ modulate Wnt signaling through TEAD-dependent and independent events and confirm that TEADi can discern YAP1/TAZ-TEAD specific effects.

- See potential effects on VGLL4 activity in response to Reviewer #3, point 3.

Based on this new information, we believe that our results confirm that the effects observed by TEADi are indeed mediated by YAP1/TAZ-TEAD activity. All new data and discussion have been added to the updated version of the paper.

2. A related concern is that the authors conclude that the observed effects of TEADi on keratinocyte differentiation, proliferation, and gene expression are due to loss of YAP/TAZ-TEAD function. Though a logical and likely conclusion, they do not rule out the possibility that TEADi promotes differentiation or KLF4 expression through a YAP/TAZ-independent mechanism. Some in vitro experiments that demonstrate that TEADi does not promote the same changes in differentiation markers, proliferation, KLF4 and some key KLF4 target genes in YAP/TAZ KD cells should address this concern. One would also expect YAP/TAZ KD to at least partially phenocopy TEADi expression. These experiments could also help address concern #1.

We understand the concern of the Reviewer. As detailed above, we performed the suggested experiments combining siYAP1/TAZ with TEADi.

3. The schematics in Figure 3H and Figure 5E and their conclusions appear to suggest that disruption of YAP/TAZ-TEAD interaction promotes YAP (TAZ?) interaction with KLF4 and the recruitment of YAP or TAZ to the promoters of KLF4-regulated genes. The authors show that KLF4 interacts with YAP, TAZ, and TEAD, and that TEADi increases this interaction (though not the TAZ-KLF4/interaction). Though consistent with their model, these data alone are not sufficient to conclude this. It remains possible that the reduced YAP/TAZ-TEAD interaction alters KLF4-mediated transcription through an indirect mechanism or through altered expression of some other co-factor. If the authors want to conclude that a YAP/KLF4 complex is driving transcription and that this is enhanced by loss of YAP-TEAD interaction, more data is necessary. For example, is YAP recruited to the promoters of KLF4 target genes and does TEADi enhance this? Does disruption of YAP/KLF4 interaction impair the expression of key KLF4 regulated genes and keratinocyte differentiation? Alternatively, the authors could discuss this as one potential model that is support by their data, but that requires additional investigation.

We agree with the concerns of the Reviewer. We added new data to confirm the mutual regulation of transcriptional networks under TEAD and KLF4 by performing RNA-seq in N/TERT2G keratinocytes with knockdown of KLF4 (siKLF4, new Supplementary Table 3 and new Fig. 3e). IPA analysis of upstream networks affected by siKLF4 indicated an activation of transcriptional targets of MITF, E2F factors and YAP1 (new Fig. 3f), showing that reduction in KLF4 levels leads to an increase transcriptional activity downstream of YAP1. Indeed, differentially regulated genes by siKLF4 showed a significant overlap with genes differentially regulated by TEADi at 48hs and by siYAP1/TAZ (new Fig. 3g), further confirming

that the transcriptional networks downstream from KLF4 and TEAD-YAP1/TAZ are closely intertwined. Since KLF4 is primarily involved in the specification of differentiated cells, we analyzed the expression of differentiation markers and found that TEAD_i and siYAP1/TAZ consistently lead to upregulation of key differentiation genes, including *IVL*, *FOXN1*, *KRT1* and *KRT10*, while siKLF4 results in downregulation of these transcripts (new Fig. 3h).

It is worth noting that siYAP1/TAZ produces an increase in KLF4 mRNA and protein levels and activation of differentiation (new Fig. 3h and 3i), indicating that YAP1/TAZ expression is not necessary for activation of differentiation and that probably YAP1/TAZ interaction with KLF4 does not have functional consequences during differentiation. Indeed, our reporter analysis with KLF4 found only marginal increases in KLF4 activity mediated by YAP1 overexpression (Supplementary Fig. 4d). Although we demonstrate that KLF4 can directly bind and regulate the activation of YAP1/TAZ-TEAD complexes, the precise mechanism by which KLF4 acts and by which TEAD regulates KLF4 expression and activity requires further investigation and we feel is beyond the scope of the current study.

We have corrected the figures and text to better accommodate the current model and highlight the limitations of our results.

4. Though the data are convincing, it is not clear the appropriate statistical tests were used. In several figures the legend suggests that a T-test was used to establish significance, yet many panels include multiple comparisons and neither the legend nor the methods indicate whether a correction for multiple comparisons was done (for example T-test with Bonferroni correction).

We thank the Reviewer for bringing this point to our attention. We have corrected the figure legends to identify more precisely the test used. In cases where 2 groups were compared, we utilized t-test. In cases where 3 to 6 groups were compared, we utilized ANOVA with multiple comparison post-test. For RNAseq data, differentially regulated genes were considered as having an absolute fold change (|FC|) of 1.5 and a false discovery rate (FDR) adjusted q-value (q) below 0.05.

Reviewer #2:

The tumor biology-related aspect of the manuscript is comparatively weak and could well be preserved for a future contribution.

We agree with the Reviewer that the tumor biology aspect of our manuscript requires additional work. We are in the process of performing a full in vitro and in vivo analysis of the effects of TEAD inhibition on skin tumor formation. However, the dissection of these mechanisms will take a significant additional amount of time. We have decided, as recommended by the Reviewer and considering the new data added to the manuscript, to remove the tumor biology-related data. We are in the process of finishing up this aspect of the study and hope that the Reviewers and the Editor agree that the analysis will best be reported in a follow up study.

This is a substantive, original, important and beautiful study that should be published. I do have comments and suggestion which I believe to be reasonable, and I am confident that the authors can and will consider and address them.

Fig 1a / How big is TEAD_i? The should be related in the main text. An effective strategy would be to provide amino acid landmarks for the various domains making up TAED_i, as shown in Fig 1a. Also, perhaps the authors should enhance their molecular account of how they think TEAD_i functions.

We agree with the Reviewer. We expanded the Results section of “Development of a genetically-encoded YAP1/TAZ-TEAD interaction inhibitor” to further include these recommendations. However, the 3D structures of the domains and points of contact of the TBDs with TEAD have been shown in detail in previous reports^{1, 2, 3}, so we consider that referring to the original publications is the most respectful way

to address this point. The amino-acid sequence for all domains and full peptide is included in the “DNA constructs” section of “Methods”.

Fig 1h / The authors should quantitate the frequency of K10 positive cells in the control and TEADi-expressing epidermal equivalents.

We agree with the Reviewer. We have included the quantification as new Supplementary Fig. 1i

Fig 2 / The authors should make available their entire RNAseq data set.

We agree with the Reviewer. All RNAseq data has been uploaded to GEO and we have also included three new Supplementary Tables with the analysis of the data. Information on how to access GEO submissions is included in the manuscript in the Methods section under Data availability.

Fig 2c / The overlap in TEAD-sensitive genes (n=4) with other/previous studies is extremely limited (4 out of a potential of >1500 ORFs – and no statistical test being reported, unlike other instances of similar comparisons elsewhere in the ms.). Why is it so limited, and is the main conclusion actually supported by the data?

We apologize for the confusion with this Figure. The figure shows the overlap of our data with two previously published datasets of YAP targets. The 4 “common” genes arise from the triple intersection and is due to a poor overlap of the Zanconato et al. and Zhao et al. datasets. Our dataset has significant overlap with the cited datasets (now shown in Fig. 2c). The figure has been corrected, visualizations updated for better clarity, and statistical analysis of intersections by hypergeometric test (one tailed Fisher's exact test) has been included. In addition, to further compare the transcriptional consequences of TEAD blockage with those triggered by loss of YAP1 and TAZ, we performed RNA-seq in N/TERT2G keratinocytes with YAP1 and TAZ knockdown by pooled-small interfering RNA. See reply to point #1 from Reviewer #1.

p 6 / The authors state: “revealing that cell cycle exit and activation of differentiation programs in keratinocytes are not concomitantly activated”. They should explore the literature for precedents, and report on them.

We agree with the reviewer. It is still not clear in the stem cell field whether the reorganization of the cell cycle represents the cause or consequence of cell differentiation ⁷. We understand that our data does not completely address this point, so we have removed the sentence.

p 6 / The authors conclude: “...and that cell cycle exit is not responsible for the activation of differentiation programs.” It appears that they don't have enough evidence to make the latter aspect of the conclusion “conclusive”.

We agree with the reviewer. We understand that our data does not completely address this point, so we have removed the sentence.

Fig 3c / The authors limit their WB analysis for differentiation product to K10 – it would be very desirable to test for additional markers if they wish to maintain a claim about a global effect on keratinocyte differentiation as opposed to a specific effect on K10 expression. Also, correct the CRY61 erroneous label in this figure.

We agree with the reviewer. Utilizing our newly generated data on siYAP1/TAZ and KLF4, we have performed a side by side analysis of all the differentially expressed differentiation markers (new Fig. 3h). We have also corrected the labelling for CYR61.

p 8 / (minor) I would refrain from using the term “hyperplasia” to describe what’s happening in these compensated epidermal areas – I would simply use the term “thickened epidermis”. Hyperplasia is usually used to convey enhanced proliferation.

We agree with the reviewer. We have replaced the term “hyperplasia” for “thickened epidermis” as suggested.

P 8 (major) I think the authors really need to provide additional evidence for their account of the cellular mechanism(s) that account for the thickened epidermis in these compensated regions. I am not aware of any precedent for this. This is a very weak spot in the manuscript.

We agree with the reviewer and we have performed further analysis in this aspect (included in the Results section): mice that developed ulcers after 10 to 20 days of TEADi induction showed thickened epidermis in several areas surrounding wounds (Fig. 4a and Fig. 5a). These thickened epidermal areas were mostly composed by cells positive for KRT10 and loricrin and negative for the basal markers p63 and KRT5 (Fig. 5a and Supplementary Fig. 5a), indicating that they are differentiated cells. One possible explanation for these thickened and differentiated epidermal areas is that cells transitioning into differentiation with low expression of TEADi can still proliferate. Indeed, we were able to detect proliferating cells in the epidermis of mice at 12 days that present low levels of KRT5 staining (Fig. 4c and new Fig. 5b). Proliferating cells were also negative for TEADi expression (new Fig. 5c) and we observed a reduction over time of cells expressing detectable levels of TEADi (new Fig. 5d), probably caused by the fact that TEADi positive cells rapidly differentiate, leading to a termination of K5rtTA expression and, consequently, TEADi downregulation. Of interest, labeling of lymphoid cells and macrophages in the skin with CD45 indicated an increase in inflammatory cell infiltration by 4 days that is further increased by 12 days following TEADi expression (new Fig. 5e). Inflammatory cell infiltration is commonly caused by the disruption of epithelial integrity and could be a plausible trigger for the observed keratinocyte activation and thickened epidermal areas at late time points.

p 9 Fig 6 / Again here the authors should show make their entire RNA seq data set available.

We agree with the Reviewer. All RNAseq data has been uploaded to GEO and we have also included three new Supplementary Tables with the analysis of the data.

p 9 Fig 6 / The authors should provide substantive evidence that SCC15 and SCC13 cells are actually capable of a significant degree of differentiation before concluding about the uncoupling of the impact of TEADi on cell cycle vs. terminal differentiation. What are their positive controls for substantive differentiation of these cell lines? This concern is further fueled by the last paragraph of the results section. I think the data supporting the maintenance of a link between KFL4 function and TAEDi is likely correct – but I would recommend a more prudent interpretation of the implications for the regulation of terminal differentiation (or lack thereof) in aberrant models such as SCc13 and SCC15...

We agree with the Reviewer that this is a weak point in our study. As mentioned above, we are in the process of performing a full in vitro and in vivo analysis of the effects of TEAD inhibition on skin tumor formation and we have removed the tumor biology-related data from the current study.

Fig 6 / The statistical p value for overlap ($p < 0.05$) is borderline.

We apologize for the confusion with this figure. The p value shown was of the selected genes in the graph. We have now included statistical analysis of overlaps by hypergeometric test (one tailed Fisher's exact test).

A prediction from the data herein reported is that KLF4 regulation and function may well be mechanosensitive in the epidermis. The authors should comment on this.

We thank the Reviewer for bringing this point to our attention. We have added this to the Discussion.

Reviewer #3:

While the manuscript has some interesting observations regarding YAP-TEAD-KLF4 interaction and has developed a genetic tool for in vivo and in vitro studies, most findings are not entirely novel and the development of TEADi is not well justified nor tested in a way that would be required to prove its advantage over knock out studies. Thus, apart from other relatively minor problems, I have three major concerns:

1. Lack of significant novelty: The observation that a TEAD dependent genetic program is required for skin homeostasis has been made before in a number of papers that have gone deeper than the current manuscript.

We agree with the Reviewer that previous studies have shown TEAD and YAP1/TAZ involvement in skin homeostasis, and such studies are cited and commented in the article. However, that is not the main point of our study. As mentioned in the introduction, we aim at dissecting the specificity of TEAD-dependent transcriptional effects downstream of YAP1 and TAZ. We have modified the text and added numerous new data and figures to highlight better the novelty of our study. It is important to note that all studies with transcriptome-wide datasets of YAP1 or TAZ targets that we found and are cited in the manuscript were done with siRNA or knockout approaches towards YAP1 or TAZ and not TEAD. These studies do not analyze the effects of specifically targeting TEAD transcription factors nor they compare the impact of targeting YAP1/TAZ versus targeting TEAD. As mentioned below in point 2, due to the fact that disruption of YAP1/TAZ-TEAD complexes has become a main druggable target, we believe that our study brings an unexplored aspect that describes the fundamental biological commonalities and differences of YAP1/TAZ depletion versus YAP1/TAZ-TEAD disruption.

We hope that the new version of the manuscript and the newly added information give an improved understanding of the novelty of our study and usefulness and advantages of the TEADi model.

2. The necessity and usefulness of the TEADi is poorly justified. Many studies have successfully used YAP;TAZ double knock out strategies, so this cannot be it. Also, the authors argue that TEADi would circumvent the problem of disrupting potential YAP/TAZ function in the cytoplasm and interactions with other transcription factors. However, fail to make a convincing case for their argument. They do not show whether TEADi and YAP;TAZ double knockout indeed give different results. Without such comparison their argument that TEADi provides an advantage over YAP;TAZ knockout is fantasy.

We understand that the Reviewer is confused regarding the usefulness of our approach. To make this clearer, we have explained in more detail our strategy and the usefulness of our system in the new version of the manuscript. TEADi is aimed as an additional resource to the field, with improved advantages that include rapid and simple inhibition of TEAD transcription and specific blockage of nuclear events mediated by both YAP1 and TAZ without affecting structural or cytoplasmic functions of these proteins. As highlighted by the Reviewer #1 “TEADi is seemingly a very powerful reagent that is long overdue and will be extremely useful to researchers in the field”. This comment as well as other comments received during presentations of our work at conferences indicate the interest in the field for a tool like the one we present in this study.

Considering that the HIPPO-pathway constitutes one of the top signaling pathways altered in human cancer ⁸, disruption of YAP1/TAZ-TEAD complexes has become a main target to suppress oncogenic activity. TEADi could be potentially used to dissect the TEAD-dependent and independent roles of YAP1/TAZ signaling and aid in the design of improved targeting strategies for this pathway in cancer and other pathologies, particularly to assess the sensitivity of tissues and tumor models to YAP1/TAZ-TEAD

disruption in vivo. To our knowledge, no other mouse model is available that can be used to disrupt YAP1/TAZ-TEAD complexes in a tissue and time-controlled manner. To demonstrate whether TEADi could be used to distinguish between YAP1/TAZ and TEAD specific events we investigated the activation of Wnt signaling in our cells. Regulation of the activation of β -catenin has been shown to be independent of TEAD transcription factors and dependent on the direct interaction of YAP1 and TAZ with components of the Wnt signaling pathway^{4,5}. Analysis of canonical pathways affected by siYAP1/TAZ and TEADi in Ingenuity Pathway Analysis (IPA) indicated that, although under both conditions Wnt/ β -catenin signaling gene networks are differentially regulated, siYAP1/TAZ leads to a clear increase in the activity of the pathway when compared with TEADi (new Fig. 2e, f and g). Our results indicate that YAP1/TAZ modulate Wnt signaling through TEAD-dependent and independent events and confirm that TEADi can discern YAP1/TAZ-TEAD specific effects.

Regarding specificity and comparison of effects of TEADi with YAP1/TAZ depletion please refer to the above answer to Reviewer #1, point #1.

3. While this study is nicely illustrated, it is shallow and in places naïve and intellectually ignorant. For example, TEADs are known to interact with VGLL4, which counteracts Yap, Taz. However, there is no mention of this let alone a test whether this interaction is affected, nor an argument how this may make TEADi different from YAP/TAZ knockout. Given that the YAP/TAZ-TEAD interaction inhibitor also inhibits Vgl4-TEAD interaction, how can these phenotypes be entirely attributed to YAP/TAZ inhibition? How do these phenotypes compare to when VGLL4 is overexpressed and knocked out in basal cells? Also, the analysis of gene expression profiles too superficial to be at a level required for a publication with impact. For example, the authors compare their profile with that of two other studies (Zhao and Zanconato). They simply say there is significant overlap. However, I find the overlap between these three studies surprisingly minimal. This is especially true between the other two published studies, where only a fraction of enriched genes overlap. I am thus surprised how the authors interpret these data, it seems to me that wishful thinking and biased expectations were dictating the argument rather than a careful analysis of the results. I thus think that the authors really missed a chance here and in other parts of their study to truly figure out something novel, rather than putting together an poorly supported argument that does not even provide any novelty.

It has been shown in numerous studies that the main function of VGLL4 in cells is to block the binding of YAP1/TAZ to TEAD transcription factors [^{1, 9, 10, 11, 12, 13, 14}, just to name a few], so TEADi as well as other published inhibitors take advantage of this fact to use the binding domain of VGLL4 as a dominant negative for TEAD activity. We do not claim that the phenotype observed is due entirely to YAP1/TAZ inhibition, we rather attribute it to inhibition of TEAD, which is what endogenous VGLL4 would also be doing. In the eventual case that TEADi would be blocking additional unknown functions of VGLL4, VGLL4 knockout mice have no reported phenotype on skin development¹⁴, indicating that blockage of VGLL4 in keratinocytes might not have any functional consequences. In addition, we have included a series of new experiments to demonstrate the specificity of TEADi towards YAP1/TAZ-TEAD signaling (please refer to the above answer to Reviewer #1, point #1). Based on all this evidence, we believe that the argument that the effects we see could be attributed to the blockage of some unknown function of VGLL4 are unsubstantiated.

Regarding the overlap with Zhao and Zanconato datasets, as detailed in the response to Reviewer #2, the 4 “common” genes arise from the triple intersection and is due to a poor overlap of the Zanconato et al. and Zhao et al. datasets. Our dataset has significant overlap with the cited datasets (now shown in Fig. 2c). The figure has been corrected, visualizations updated for better clarity, and statistical analysis of intersections by hypergeometric test (one tailed Fisher's exact test) have been included. This analysis is

presented to support the specificity of our inhibitor and is meant to highlight potential conserved targets of YAP1/TAZ-TEAD among different cell types and studies.

Other:

How does KLF4 regulate the YAP-TEAD mediated CYR61 expression? Does KLF4 overexpression or KD affect YAP-TEAD binding? Co-IP of YAP-TEAD in presence/absence of KLF4 and its repressor/activator variants (1-566 and 159-329).

As shown in Fig. 3j, KLF4 binds to the YAP1/TAZ-TEAD complex and KLF4 knockdown affects YAP1-TEAD binding (Fig. 3d). While the mechanism by which this binding causes repression of TEAD transcription is not clear, we find that both the repressor and activator domains of KLF4 can reduce TEAD transcriptional activity (new Supplementary Fig. 4e and f) and can interact with TEAD, particularly the activation domain of KLF4 (new Fig. 3k).

As mentioned in the response to point #3 from Reviewer #2, although we demonstrate that KLF4 can directly bind and regulate the activation of YAP1/TAZ-TEAD complexes, the precise mechanism by which KLF4 acts and by which TEAD regulates KLF4 expression and activity requires further investigation and we feel is beyond the scope of the current study. We have corrected the figures and text to better accommodate the current model and highlight the limitations of our results.

Does dox inducible TEADi reduced basal cell proliferation or caused their elimination? Are basal cells positive for TUNEL and cCasp3?

We thank the Reviewer for bringing this point to our attention. As shown in Fig. 1h TEADi induces expression of KRT10 in basal cells and reduces the amount of PCNA+ proliferative cells (quantification in Supplementary Fig. 1 h and i). We have included new data showing that TEADi does not induce expression of apoptosis markers in keratinocytes (new Supplementary Fig. 2c), indicating that the effects are mainly mediated by regulation of proliferation and differentiation.

What happens with cell proliferation and cell death in the epidermis reconstruction assay stably expressing a tetracycline-inducible TEADi (Fig1G)?

We did not find cCasp3 positive cells in the epidermis reconstruction assays. As mentioned in the previous point, we have included new data showing that TEADi does not induce expression of apoptosis markers in keratinocytes (new Supplementary Fig. 2c).

TEADi induction caused epidermal hyperplasia around wounds (Fig. 4b and 4c). Are these hyperplastic areas expressing TEADi and thus negative for YAP activity? How could these YAP repressed cells proliferate, but basal cells show reduced proliferation? In particular, if the authors showed that TEADi leads to a downregulation of cell cycle processes in both cancer and normal cells, how could a tissue become hyperplastic?

We have performed further analysis in this aspect, please see response to point #8 from Reviewer #2.

References

1. Jiao S, *et al.* A peptide mimicking VGLL4 function acts as a YAP antagonist therapy against gastric cancer. *Cancer Cell* **25**, 166-180 (2014).
2. Zhang Z, *et al.* Structure-Based Design and Synthesis of Potent Cyclic Peptides Inhibiting the YAP-TEAD Protein-Protein Interaction. *ACS Med Chem Lett* **5**, 993-998 (2014).
3. Hau JC, *et al.* The TEAD4-YAP/TAZ protein-protein interaction: expected similarities and unexpected differences. *Chembiochem* **14**, 1218-1225 (2013).

4. Barry ER, *et al.* Restriction of intestinal stem cell expansion and the regenerative response by YAP. *Nature* **493**, 106-110 (2013).
5. Azzolin L, *et al.* YAP/TAZ incorporation in the beta-catenin destruction complex orchestrates the Wnt response. *Cell* **158**, 157-170 (2014).
6. Sakanaka C. Phosphorylation and regulation of beta-catenin by casein kinase I epsilon. *J Biochem* **132**, 697-703 (2002).
7. Liu L, Michowski W, Kolodziejczyk A, Sicinski P. The cell cycle in stem cell proliferation, pluripotency and differentiation. *Nature cell biology* **21**, 1060-1067 (2019).
8. Sanchez-Vega F, *et al.* Oncogenic Signaling Pathways in The Cancer Genome Atlas. *Cell* **173**, 321-337 e310 (2018).
9. Vaudin P, Delanoue R, Davidson I, Silber J, Zider A. TONDU (TDU), a novel human protein related to the product of vestigial (vg) gene of *Drosophila melanogaster* interacts with vertebrate TEF factors and substitutes for Vg function in wing formation. *Development* **126**, 4807-4816 (1999).
10. Guo T, *et al.* A novel partner of Scalloped regulates Hippo signaling via antagonizing Scalloped-Yorkie activity. *Cell Res* **23**, 1201-1214 (2013).
11. Simon E, Faucheux C, Zider A, Theze N, Thiebaud P. From vestigial to vestigial-like: the *Drosophila* gene that has taken wing. *Dev Genes Evol* **226**, 297-315 (2016).
12. Zhang W, *et al.* VGLL4 functions as a new tumor suppressor in lung cancer by negatively regulating the YAP-TEAD transcriptional complex. *Cell Res* **24**, 331-343 (2014).
13. Zhang Y, *et al.* VGLL4 Selectively Represses YAP-Dependent Gene Induction and Tumorigenic Phenotypes in Breast Cancer. *Scientific reports* **7**, 6190 (2017).
14. Yu W, *et al.* VGLL4 plays a critical role in heart valve development and homeostasis. *PLoS Genet* **15**, e1007977 (2019).

Reviewers' Comments:

Reviewer #1:

Remarks to the Author:

The authors have addressed my concerns and I support publication of this very nice manuscript.

Reviewer #2:

Remarks to the Author:

The authors have made substantive revisions to their manuscript, which is improved and now suitable for publication in the view of this reviewer ..

Reviewer #3:

Remarks to the Author:

The authors present some more data. However, their response shows that they do not understand the logic of the main concerns from me (and reviewer #1) regarding the argument that TEADi has effects that are not explained by Yap/Taz/TEAD inhibition.

The question that needs to be addressed is does TEADi recapitulate the phenotype of Yap/Taz knockout or not? If the answer would be yes then the use of TEADi could indeed be used to study the functions of Yap/Taz. However, the answer is no, and thus the obvious question is why not? There are several possibilities that need to be distinguished: 1) Is TEAD required for more than mediating the activity of Yap/Taz? 2) Do Yap/Taz do more than regulate gene expression by binding to TEAD? and/or, does TEADi cause artifacts that have nothing to do with Yap/Taz/TEAD function?

It seems that the authors do not understand this problem and do not address whether or not TEADi causes artifacts. They do show some RNAseq data but these are poorly analyzed and basically not incorporated into the paper. Without explicit argumentation and data to address all three possibilities, this study is not worth much (although it will be cited and used for other studies). I agree that TEADi may indeed provide an interesting tool, however at the moment this is just an assumption and not supported by data. Of course, TEADi can already be used for experiments but the biological meaning of the results is unknown.

Two main concerns that need to be addressed:

1. The authors need to do RNAseq on the Yap/Taz knockdown plus TEADi cells. Does TEADi cause additional effects and what does that mean?

2. The authors present data showing that 25% of all protein coding genes are differentially regulated in the Yap/Taz knockdown cells. This is hardly meaningful. Similarly, TEADi is shown to cause large shifts in gene expression. It may thus not be a surprise that there is some "overlap" in gene expression. The authors need to expand their statistical analysis and use more stringent parameters (compare top 500 differentially regulated genes for example) and also include GSEA (also of only the top differentially expressed genes). The way it is presented now is rather meaningless. Also, what about other Yap signatures that are maybe better suited than the two they used?

Other:

3. The authors write "our results confirm that the effects observed by TEADi are indeed mediated by YAP1/TAZ-TEAD activity." This statement illustrates that the authors are confused: TEADi may

inhibit Yap/Taz but the effects of TEADi are not be "mediated" by Yap/Taz (which would mean that Yap/Taz are required for TEADi function, the opposite of what is meant). Also, do they mean here that TEADi indeed inhibits Yap/Taz? I thought they observe that Yap/Taz have TEAD independent effects and also that TEADi affects genes that are not regulated by Yap/Taz? Statements like this are just too imprecise and ignorant and miss interesting points of their study.

4. Similarly: "Inhibition of TEAD transcription in keratinocytes and mouse skin revealed that YAP1/TAZ-TEAD regulate epithelial cell homeostasis at two levels:" This is only true if TEADi precisely recapitulates the Yap/Taz mutant phenotype.

5. By utilizing an antibody that recognizes non-phospho active β -Catenin (Ser45) 27, we confirmed that siYAP1/TAZ results in an increase in the amount of active β -catenin in keratinocytes, while TEADi results in minimal alterations in the levels of active protein (Fig. 2g).

To me it seems that there is a minimal difference actually! It seems that the authors just writing what they wish to be true and do not properly analyze and represent the data. Thus, the data is wrongly interpreted without presenting quantification.

NCOMMS-19-16447A Response to Reviewers' comments:

Reviewers comments are in italic

Reviewer #1:

The authors have addressed my concerns and I support publication of this very nice manuscript.

Reviewer #2:

The authors have made substantive revisions to their manuscript, which is improved and now suitable for publication in the view of this reviewer.

We are pleased that Reviewers #1 and #2 find the manuscript appropriate for publication.

Reviewer #3:

The authors present some more data. However, their response shows that they do not understand the logic of the main concerns from me (and reviewer #1) regarding the argument that TEADi has effects that are not explained by Yap/Taz/TEAD inhibition.

The question that needs to be addressed is does TEADi recapitulate the phenotype of Yap/Taz knockout or not? If the answer would be yes then the use of TEADi could indeed be used to study the functions of Yap/Taz. However, the answer is no, and thus the obvious question is why not? There are several possibilities that need to be distinguished: 1) Is TEAD required for more than mediating the activity of Yap/Taz? 2) Do Yap/Taz do more than regulate gene expression by binding to TEAD? and/or, does TEADi cause artifacts that have nothing to do with Yap/Taz/TEAD function?

It seems that the authors do not understand this problem and do not address whether or not TEADi causes artifacts. They do show some RNAseq data but these are poorly analyzed and basically not incorporated into the paper. Without explicit argumentation and data to address all three possibilities, this study is not worth much (although it will be cited and used for other studies). I agree that TEADi may indeed provide an interesting tool, however at the moment this is just an assumption and not supported by data. Of course, TEADi can already be used for experiments but the biological meaning of the results is unknown.

We understand that despite our efforts and extensive changes and new experiments added to the manuscript some concerns remain regarding possible additional effects of TEADi and we thank the Reviewer for raising this concern and giving us the opportunity to clarify this further.

Regarding the questions:

1) Does TEADi recapitulate the phenotype of YAP1/TAZ knockdown?

Our data shows that indeed TEADi and siYAP1/TAZ both lead to similar effects (Fig. 2d, 2e, 3g, 3h, 3i, and Supplementary Fig. 2a, 2d, 2e), particularly in the increase in keratinocyte differentiation and activation of KLF4. We have added new data that shows that siYAP1/TAZ also decreases keratinocyte proliferation (new Supplementary Fig. 2b), indicating that TEADi recapitulates the proliferation and differentiation phenotype of YAP1/TAZ knockdown. However, an interesting and expected aspect of our study is that the effects of blocking TEAD vs knocking down YAP1 and TAZ are not completely overlapping (Fig. 3d, 3e, 3f, 3g, and Supplementary Fig. 3c, 3d and 3e). Differences in TEADi vs siYAP1/TAZ might be due to additional compensatory mechanisms that are differentially activated in conditions in which YAP1 and TAZ are absent, conditions that are not the same when TEAD binding is blocked and YAP1 and TAZ remain present to interact with other factors; timing of a protein inhibitor (TEADi) versus a gradual decrease in protein expression (siRNA); and the fact that siRNA does not completely deplete YAP1 protein expression (Supplementary Fig. 2a). Indeed, we show as an example that Wnt signaling is differentially regulated under these two conditions. We have included diagrams to better illustrate the main conclusions of our study and the differences between TEADi and siYAP1/TAZ (New Supplementary Figures 6a and 6b).

2) Is TEAD required for more than mediating the activity of YAP1/TAZ?

To our knowledge, no endogenous function for TEAD has been reported that is completely independent of YAP1 and TAZ. Basal TEAD activity is minimal in cells where YAP1 and TAZ are not active and the presence of these co-factors is necessary for TEAD activity^{1, 2, 3, 4, 5}. Nevertheless, we agree with the reviewer that potential YAP1 and TAZ independent functions could be affected and have updated the discussion to better highlight this possibility (see point 3 in other comments below).

3) Do YAP1/TAZ do more than regulate gene expression by binding to TEAD?

As mentioned in the paper, YAP1/TAZ-TEAD complexes can modulate transcription at several levels that include direct promoter regulation, enhancer association, or binding with chromatin regulating proteins^{6, 7, 8}, indicating that TEAD can

control epithelial homeostasis by direct promoter-regulation of gene expression and indirectly by modulating chromatin accessibility of other transcription factors. All these will ultimately lead to regulation of gene expression.

4) Does TEADi cause artifacts that have nothing to do with YAP1/TAZ-TEAD function?

As with any other model system, we cannot completely rule out possible non-specific effects. However, our data shows that the main effects of TEADi are recapitulated by knockdown of YAP1/TAZ. This, and numerous other studies of the specificity of the TBDs used, indicate that most of the effects observed by TEADi are mediated by YAP1/TAZ-TEAD activity and, as explained above, TEADi and siYAP1/TAZ will not cause completely overlapping results. We agree with the Reviewer that, despite the evidence presented, unexpected potential off-target effects of TEADi are possible and we have highlighted the limitations of our model in the discussion.

As we mentioned before, our system is not aimed as a replacement for YAP1/TAZ knockdown or knockout, but as an additional tool to help analyze the transcriptional effects mediated by these proteins. We have highlighted the limitations of our model in the manuscript and we are confident that additional studies will be able to further validate the usefulness of TEADi. Finally, we want to point out that Reviewer #1, which had similar concerns, indicated: *“the authors have addressed my concerns and I support publication of this very nice manuscript”*.

Two main concerns that need to be addressed:

1. The authors need to do RNAseq on the Yap/Taz knockdown plus TEADi cells. Does TEADi cause additional effects and what does that mean?

We understand the concern of the reviewer. However, we need to mention that this experiment was suggested by Reviewer#1 in the previous version of the manuscript. As suggested by Reviewer #1, we performed control experiments combining TEADi with siYAP1/TAZ to demonstrate that the main effects of induction of differentiation and KLF4 are indeed triggered by YAP1/TAZ and that these effects are not altered by further blockage by TEADi. In addition, we performed a side by side analysis of TEADi with siYAP1/TAZ to demonstrate that the main effects of TEADi are recapitulated by knockdown of YAP1 and TAZ and we have added new data that shows that siYAP1/TAZ also decreases keratinocyte proliferation (new Supplementary Fig.2b), indicating that TEADi recapitulates the proliferation and differentiation phenotype of YAP1/TAZ knockdown (see above for an explanation of additional non-overlapping effects observed in TEADi vs siYAP1/TAZ). Reviewer #1 considered our answers and experiments to appropriately address the question of TEADi specificity and it is not clearly stated why Reviewer#3 does not consider our control experiments valid. As mentioned before, we agree with the Reviewer that unexpected potential off-target effects of TEADi are possible (like in any other model system) and we have highlighted the limitations of our model in the discussion.

2. The authors present data showing that 25% of all protein coding genes are differentially regulated in the Yap/Taz knockdown cells. This is hardly meaningful. Similarly, TEADi is shown to cause large shifts in gene expression. It may thus not be a surprise that there is some "overlap" in gene expression. The authors need to expand their statistical analysis and use more stringent parameters (compare top 500 differentially regulated genes for example) and also include GSEA (also of only the top differentially expressed genes). The way it is presented now is rather meaningless. Also, what about other Yap signatures that are maybe better suited than the two they used?

We understand the concern of the Reviewer that the high number of significantly altered genes in the siYAP1/TAZ dataset can lead to confusing comparisons. This high number of genes is due to the fact that we used the same cutoff levels for significant genes for all datasets to apply an unbiased approach. In order to demonstrate that overlaps among datasets are not due to chance, we employ the commonly used hypergeometric test (one tailed Fisher's exact test), which calculates a p-value to determine the probability that the association between the focus genes in different datasets is explained by chance alone. This p-value is included in all analysis of overlaps and is highly significant for the siYAP1/TAZ and TEADi intersections ($p=5.491e-20$ for 24hs and $p=5.169e-79$ for 48hs, Fig. 2d). To further address this concern, we have included new analysis of the dataset using more stringent parameters as suggested by the Reviewer, including gene set enrichment analysis by gene ontology (GO) (new Supplementary Fig. 2e).

The Zanconato et al 2015 and Zhao et al 2008 dataset were selected for comparison since they are well accepted and widely used signatures for YAP1 targets in mammalian cells.

Other:

3. The authors write "our results confirm that the effects observed by TEADi are indeed mediated by YAP1/TAZ-TEAD activity." This statement illustrates that the authors are confused: TEADi may inhibit Yap/Taz but the effects of TEADi are not be "mediated" by Yap/Taz (which would mean that Yap/Taz are required for TEADi function, the opposite of what is meant). Also, do they mean here that TEADi indeed inhibits Yap/Taz? I thought they observe that Yap/Taz have TEAD

independent effects and also that TEAD_i affects genes that are not regulated by Yap/Taz? Statements like this are just too imprecise and ignorant and miss interesting points of their study.

We understand the concern of the reviewer and we have included additional discussion into this paragraph (changes are underlined):

“The peptides used to build TEAD_i were selected due to their proven specificity towards TEAD binding and our results suggest that the main effects of TEAD_i are indeed mediated by YAP1/TAZ-TEAD activity. One limitation of our model is that despite the specificity of the TBDs they could cause additional nuclear effects that are not mediated by YAP1/TAZ inhibition, particularly the TBD of VGLL4. It has been shown in numerous studies that the main function of VGLL4 in cells is to block the binding of YAP1/TAZ to TEAD transcription factors, indicating that TBD-VGLL4 should not have any additional effects than to potentiate endogenous VGLL4-mediated TEAD inhibition. In addition, VGLL4 knockout mice have no reported phenotype on skin development, suggesting that blockage of VGLL4 in keratinocytes might not have any functional consequences. It is worth noting also that TEAD_i and YAP1/TAZ knockdown will cause additional non-overlapping effects due to the fact that both are different conditions (Supplementary Fig. 6b): in TEAD_i expressing cells, YAP1 and TAZ remain present to interact with other factors; while in siYAP1/TAZ cells, structural interactions of YAP1 and TAZ, including interactions with the β -Catenin destruction complex, are altered, causing additional effects not present in TEAD_i cells.”

4. Similarly: "Inhibition of TEAD transcription in keratinocytes and mouse skin revealed that YAP1/TAZ-TEAD regulate epithelial cell homeostasis at two levels:" This is only true if TEAD_i precisely recapitulates the Yap/Taz mutant phenotype. We agree with the reviewer and we have corrected the sentence to "Inhibition of TEAD transcription in keratinocytes and mouse skin revealed that TEAD activity regulates epithelial cell homeostasis at two levels:".

5. By utilizing an antibody that recognizes non-phospho active β -Catenin (Ser45) 27, we confirmed that siYAP1/TAZ results in an increase in the amount of active β -catenin in keratinocytes, while TEAD_i results in minimal alterations in the levels of active protein (Fig. 2g).

To me it seems that there is a minimal difference actually! It seems that the authors just writing what they wish to be true and do not properly analyze and represent the data. Thus, the data is wrongly interpreted without presenting quantification. We understand the concern of the reviewer. Our conclusion is based on several different pieces of evidence: A) IPA analysis indicating a significant alteration of classic Wnt/ β -catenin signaling gene networks ($p=0.012$ for siYAP1/TAZ and $p=0.0013$ for TEAD_i, Fig. 2e) with an overall activation Z-Score of 3.13 for siYAP1/TAZ and 0.577 for TEAD_i (Fig. 2f), suggesting that knockdown of siYAP1/TAZ leads to a more substantial activation of the pathway than TEAD-blockage which is supported by previous reports by Barry ER et al (2013) and Azzolin L et al (2014) that indicate that YAP1 and TAZ can regulate Wnt signaling in a TEAD-independent manner; and B) analysis by western blot of the levels of non-phospho active β -Catenin showing that siYAP1/TAZ leads to an increase in dephosphorylated β -Catenin (Fig. 2g). To make this result clearer and as suggested by the Reviewer, we have included quantification of independent experiments showing a significant increase in active dephosphorylated β -Catenin by siYAP1/TAZ but not TEAD_i (new Supplementary Fig. 2c).

References:

1. Ishiji T, et al. Transcriptional enhancer factor (TEF)-1 and its cell-specific co-activator activate human papillomavirus-16 E6 and E7 oncogene transcription in keratinocytes and cervical carcinoma cells. *EMBO J* **11**, 2271-2281 (1992).
2. Zhao B, et al. TEAD mediates YAP-dependent gene induction and growth control. *Gene Dev* **22**, 1962-1971 (2008).
3. Zhang H, et al. TEAD transcription factors mediate the function of TAZ in cell growth and epithelial-mesenchymal transition. *The Journal of biological chemistry* **284**, 13355-13362 (2009).
4. Liu-Chittenden Y, et al. Genetic and pharmacological disruption of the TEAD-YAP complex suppresses the oncogenic activity of YAP. *Gene Dev* **26**, 1300-1305 (2012).
5. Walko G, et al. A genome-wide screen identifies YAP/WBP2 interplay conferring growth advantage on human epidermal stem cells. *Nat Commun* **8**, 14744 (2017).
6. Chang L, et al. The SWI/SNF complex is a mechanoregulated inhibitor of YAP and TAZ. *Nature* **563**, 265-269 (2018).
7. Zanconato F, et al. Transcriptional addiction in cancer cells is mediated by YAP/TAZ through BRD4. *Nature medicine* **24**, 1599-1610 (2018).
8. Zanconato F, et al. Genome-wide association between YAP/TAZ/TEAD and AP-1 at enhancers drives oncogenic growth. *Nature cell biology* **17**, 1218-1227 (2015).

Reviewers' Comments:

Reviewer #3:

Remarks to the Author:

My concerns have been addressed and arguments and interpretations are now presented with more balance including potential pitfalls and problems.

NCOMMS-19-16447A Response to Reviewers' comments:

Reviewers comments are in italic

Reviewer #3 (Remarks to the Author):

My concerns have been addressed and arguments and interpretations are now presented with more balance including potential pitfalls and problems.

We are pleased that we were able to address the concerns from the Reviewer.